# Multi-decadal geomorphic changes of a low-angle valley glacier in East Kunlun Mountains: remote sensing observations and detachment hazard assessment

Xiaowen Wang[1, 2], Lin Liu[3], Yan Hu[3], Tonghua Wu[4], Lin Zhao[4, 5], Qiao Liu[6], Rui Zhang[1, 2], Bo Zhang[1], Guoxiang Liu[1, 2]

[1]Faculty of Geosciences and Environmental Engineering, Southwest Jiaotong University, China

[2]State-Province Joint Engineering Laboratory of Spatial Information Technology of High-speed Rail Safety, Southwest Jiaotong University, China

[3]Earth System Science Programme, Faculty of Science, The Chinese University of Hong Kong, China

[4]Cryosphere Research Station on the Qinghai-Tibet Plateau, State Key Laboratory of Cryosphere Science, Northwest Institute of Eco-Environment and Resources, Chinese Academy of Sciences, China

[5]School of Geographical Sciences, Nanjing University of Information Science and Technology, China

[6]Institute of Mountain Hazards and Environment, Chinese Academy of Sciences, China

*Correspondence to*: Xiaowen Wang (insarwxw@swjtu.edu.cn)

**Abstract:** Detachments of large parts of low-angle mountain glaciers in recent years have raised great attention due to their threats to lives and properties downstream. While current studies have mainly focused on post-event analysis, few opportunities have presented themselves to assess the potential hazards of a glacier prone to detachment. Here we presented a comprehensive analysis of the dynamics and runout hazard of a low-angle (~20°) valley glacier, close to the Qinghai-Tibet railway and highway, in the East Kunlun Mountains on the Qinghai-Tibet Plateau. The changes in morphology, terminus position, and surface elevation of the glacier between 1975 and 2021 were characterized with a stereo-image pair from the historical KH-9 spy satellite, six Digital Elevation Models (DEMs), and eleven high-resolution images from Planet Labs. The surface flow velocities of the glacier tongue between 2009 and 2020 were also tracked based on cross-correlation of Planet images. Our observations show that the glacier snout has been progressively advancing in the past four decades, with a stepwise increase of advance velocity from $4.55\pm0.46$ m·a$^{-1}$ between 1975 and 2009 to $30.88\pm2.36$ m·a$^{-1}$ between 2015 and 2020. DEM differencing confirms the glacial advance, with surface thinning in the source region and thickening in the tongue. The net volume loss over the glacier tongue was about $11.21\pm2.66 \times10^5$ m$^3$ during 1975–2018. Image cross-correlation reveals that the surface flow velocity of the glacier tongue has been increasing in recent years, with the mean velocity below 4800 m more than tripled from $6.3\pm1.8$ m·a$^{-1}$ during 2009–2010 to $22.3\pm3.2$ m·a$^{-1}$ during 2019–2020. With a combined analysis of the geomorphic, climatic, and hydrologic conditions of the glacier, we suggest that the flow of the glacier tongue is mainly controlled by the glacier geometry, while the presence of an ice-dammed lake and a supraglacial pond implies a hydrological influence as well. Taking the whole glacier and glacier tongue as two endmember avalanche sources, we assessed the potential runout distances of these two scenarios using the angle of reach and the Voellmy-Salm avalanche model. The assessments show that the avalanche of the whole glacier would easily travel a distance threatening the safety of the railway. In contrast, the detachment of the glacier tongue would threaten the railway only with a small angle of reach or when employing a low friction parameter in the Voellmy-Salm modeling.

# 1. Introduction

Glacier instabilities in the form of ice break-offs and avalanches are universal phenomena (Faillettaz et al., 2015; Haeberli et al., 2004; Jacquemart et al., 2020). Most of the glacier instabilities occur on steep glacier termini or hanging glaciers, while recent studies show that ice detachment can also occur in low-angle (lower than around 20°) valley glaciers (Kääb et al., 2021a). The catastrophic detachment of part or even a whole glacier can transport ice mass downstream to a distance of more than ten kilometers, with a typical volume of $10^6$–$10^7$ m$^3$. Due to the hazardous threats of glacier detachment to people's lives and infrastructure, distinguishing the detachment -prone glaciers from the non-threatening ones is crucial for hazard mitigation.

Several extraordinary low-angle glacier detachment events have been reported in recent years. One of the earliest events that was documented in detail was the destructive 2002 Kolka glacier detachment in Russia, which killed about 140 people due to the mass flow (Haeberli et al., 2004; Huggel et al., 2005). Another destructive event raising great attention was the 2016 detachments of two valley glaciers in the Aru mountain range in the western Qinghai-Tibet Plateau (QTP), which caused nine casualties (Bai and He, 2020; Gilbert et al., 2018; Kääb et al., 2018; Tian et al., 2016). In addition to these two well-known events, a few historical detachments of valley glaciers are recently recognized and analyzed, such as the 2007 detachment of Leñas glacier in the Argentinian Andes (Falaschi et al., 2019), the three repeat detachments (in 2004, 2007, and 2016) of a glacier in the Amney Machen mountain range of the eastern QTP (Paul, 2019), and the two (2013 and 2015) detachments of the Flat Creek glacier in Alaska (Jacquemart et al., 2020). Recent research suggests that glacier detachments occur more frequently than previously thought (Kääb et al., 2021a).

A wide range of triggers can lead to a glacier detachment. Possible triggering factors include changes in ice thermal regime, morphology of a glacier, and atmospheric conditions (Gilbert et al., 2018; Tian et al., 2016, Kääb et al., 2021a). Some detachments occurred on surge-type glaciers. For example, the Kolka glacier and the Amney Machen glacier have experienced repeated surging in history (Kotlyakov, 2004; Paul, 2019). The detachments of several glaciers such as the Aru (Aru-1 & Aru-2) and Flat Creek glaciers were also preceded by geometric changes in the form of surge-like behaviors, although they were not known as surging before (Gilbert et al., 2018; Jacquemart et al., 2020). Meanwhile, studies have highlighted that sudden detachments can occur on glaciers with no historical records of instability (Kääb et al., 2018; 2020).

The detachment of Aru-1 and Aru-2 glaciers on the QTP has raised concerns on the stability of glaciers there, especially under intense climate warming. In past decades, air temperatures recorded by weather stations on the QTP have been increasing at a mean rate of 0.3~0.4 °C·10a$^{-1}$, which is twice the mean global rate (Chen et al., 2015). Considering that large-volume ice detachments can occur on low-angle mountain glaciers, it is essential to investigate the long-term dynamics of glaciers prone to detachments and further assess their potential impacts. While previous studies of glacier ice avalanches and surge movements on the QTP mainly focused on the Karakorum and West Kunlun mountain regions where a large number of surge-type glaciers exist (Bhambri et al., 2020; Yasuda and Furuya, 2015), little is known about glacier instabilities in the inner region of the plateau.

In this study, we present a comprehensive analysis of the dynamics of a small low-angle valley glacier (94.145°E, 37.678°N) in the East Kunlun Mountains of QTP (Fig. 1). We refer to the glacier's name as 'KLP-37' since it is located at the Kunlun Pass (KLP) of the East Kunlun Mountains and numbered 532EB037 in China's second glacier inventory (Guo et al., 2015). We identified the glacier, which is close to the Qinghai Tibet railway and highway, during a field trip in the KLP region in 2016 (Wang et al., 2020). The presence of intense crevassing on the glacier surface raised the question whether a hazardous ice avalanche might be imminent.

To assess the stability of the KLP-37 glacier, we employed multi-sensor satellite imagery to characterize its morphological changes and dynamics in the past 40 years. The changes in the terminus position of the glacier were tracked by interpreting optical images from the Planet constellation and Google Earth. We used six DEMs over the glacier between 1975 and 2018 to quantify the surface elevation variations. The 1975 DEM was reconstructed using a stereo image pair from the Hexagon KH-9 reconnaissance satellite. Surface flow velocities of the glacier tongue from 2009 to 2020 were also mapped using image cross-correlation. Combining the decadal geomorphic changes and surface velocities, we discuss the possible mechanisms accounting for the dynamics of the KLP-37 glacier and estimate the potential runout distance if a failure of the glacier tongue occurred in the future. We also discuss how the site-specific study on the KLP-37 glacier could provide new insights into the glacier detachment hazard monitoring and assessment.

## 2. Study site and remote sensing data

### 2.1 Study site

The KLP is located in the central part of the East Kunlun Mountains in the inner QTP (Fig. 1a). The geomorphic pattern in the KLP region was influenced by tectonic movement 1.1–0.6 Ma (i.e., Maga-annum) BP. During the maximum Quaternary (i.e., Wangkun Glaciation) glaciation 0.7–0.5 Ma in the region, the glaciers were 3–5 times larger than today (Wu et al., 2001). The KLP-37 glacier rests on the northern slope of the East Kunlun Mountains, next to the Xidatan basin. The Xidatan basin is filled with Pliocene and Quaternary alluvial, lacustrine and glacial deposits, which unconformably overlie Triassic metamorphic basement rocks (Song et al., 2005). The lithology on the slope where the glacier lies is mixed breccia, a sequence of glacial deposits termed as Wangkun till, which is composed of angular, poorly sorted gravels of slate, meta-sandstones, and mudstones (Song et al., 2005). The rock surface is highly weathered, resulting in rock fragments in a fine-grained (pelitic to sandy) matrix (Wu et al., 1982), as evidenced by the fine debris cover on the glacier tongue, suggesting the glacier likely rests on a soft bed.

The KLP region nowadays is also tectonically active. The Kunlun fault, one of the principal left-lateral strike-slip fault systems in the northern part of the QTP, runs for about 1600 km along the west-east and splays into two sub-fault segments at the KLP (Fig. 1a). The Kunlun fault system generated a few Mw>7 earthquakes in the last 100 years, including the most recent two in 1963 (Mw7.1) and 2001 (Mw7.8) (Lasserre et al., 2005). Notably, the 2001 event, with its epicenter only ~45 km west of the KLP-37 glacier (Fig. 1a), induced several ice avalanche events (see the yellow dots in Fig. 1a) over the glaciers in the KLP region (Jerome et al., 2004).

The KLP-37 glacier terminates about 3.3 km from the Qinghai-Tibet railway, which crosses the Xidatan basin along the west-east direction (Fig. 1b). The Xidatan basin is also the northern permafrost boundary of the QTP. The climate is typically cold and arid: the annual mean air temperature is -2.9 ℃ at the elevation of about 4400 m, and the average annual precipitation is about 400 mm, with most of the precipitation concentrating in summer from May to September (Luo et al., 2018; Wu et al., 2005). The snow line in this region is about 5100 m on the north slope and 5300 m on the south (Wu et al., 2001).

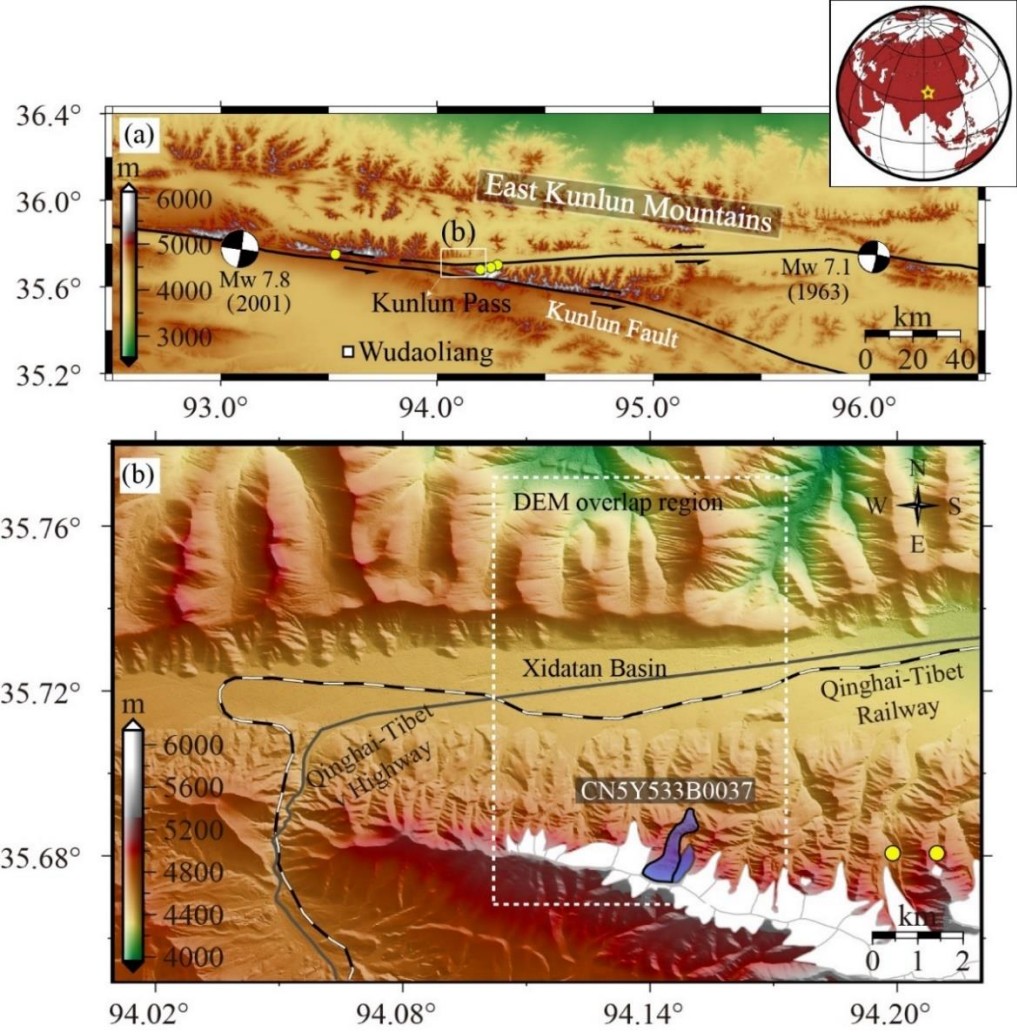

**Figure 1: (a) The tectonic overview of the study area. The black lines depict the Kunlun fault traces. The yellow dots mark the ice avalanche locations induced by the 2001 Mw 7.8 earthquake (Jerome et al., 2004). The white square indicates the Wudaoliang Meteorological station. (b) The topography of the Kunlun Pass region. The white rectangle shows the overlap region of the DEMs used in this study. The polygons are the glaciers from the GAMDAM glacier inventory (Sakai et al., 2019), while the one filled with purple color is the KLP-37 glacier, whose west branch is outlined in thick black.**

The KLP-37 glacier is a small low-angle valley glacier consisting of two branches (the GAMDAM glacier inventory, Sakai, 2019). Here our analyses mainly focus on the west branch, which has a length of about 2.06 km and a mean slope of about 20°. The elevation range of the glacier spans from 4650 to 5450 m. Compared to the east branch of the glacier, the west branch's terminus lies about 220 m lower (Fig. 2a). The width of the glacier gradually narrows from the accumulation zone to the downstream. From the satellite image taken in the summer of 2013 (Fig. 2a), we can observe an ice-dammed glacier lake in front of the glacier's east branch and a supraglacial pond on the west branch's tongue surface. The field photo on 27 June 2016 exhibits multiple horizontally distributed fissures at the glacier accumulation area (Fig. 2b). The developed crevasses, exposed ice cliffs, and glacier ice are also clearly visible in the tongue region, indicating the tongue is highly active (Fig. 2c).

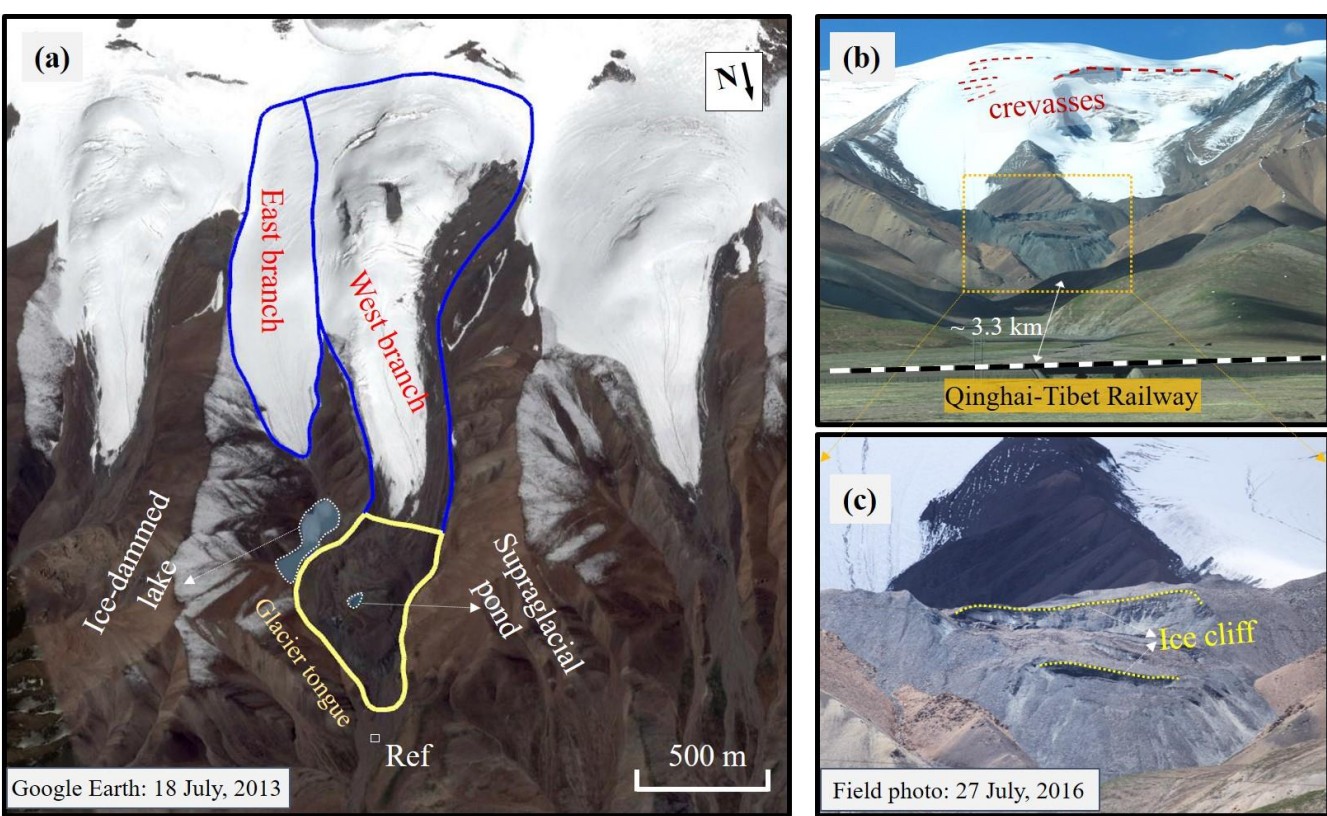

**Figure 2: (a) The outlines of the KLP-37 glacier (blue line) and tongue (yellow line) overlayed on a Google Earth image of 18 July 2013 (© Google Earth™). The ice-dammed glacier lake and a supraglacial pond are also shown. (b) and (c) are the field photos acquired by the author on 27 July 2016, which exhibit the crevasses and ice cliffs on the glacier surface. The rectangle annotated with "Ref" in (a) indicates the area for estimating image cross-correlation error (see Section 3.3).**

## 2.2 Remote sensing data

Table 1 lists the satellite remote sensing data used in this study. The data types include optical orthoimages, stereo images, and multiple DEMs derived from both stereo-photogrammetry and SAR interferometry. All the data were referenced to the coordinate system of UTM zone 46 N.

**Table 1. Satellite imagery and DEMs used in this study.**

| Data | Date | Spatial Resolution (m) | Data type | Purpose |
|---|---|---|---|---|
| RapidEye | 2009/08/30; 2010/09/07; 2012/09/12; 2013/08/04; 2015/10/12; | 5 | Orthoimages | Geomorphic change Terminus advance Flow velocity |
| PlanetScope | 2016/09/10; 2017/08/09; 2018/07/24; 2019/08/06; 2020/08/29; 2021/03/23; | 3 | Orthoimages | |
| Google Earth | 2013/08/15 | ~ 1 | - | Geomorphic change |
| KH-9 | 1975/12/10 | 7.6 | Stereo images | Terminus advance Elevation change |
| ASTER | 2018/01/12 | 15 | Stereo images | Elevation change |
| SRTM-X/C | 2000/02 | 30 | DEM | Elevation change |
| HMA DEM | 2010/12/05; 2014/08/27 | 8 | DEM | Elevation change |
| TanDEM-X DEM | 2011–2014 | 12 | DEM | GCP elevation Elevation reference |

     Glaciers changes are often mapped on the freely available Landsat and ASTER satellite images, which have a resolution of

10–30 m (Bolch et al., 2011; Scherler et al., 2011). Considering that KLP-37 is a small valley glacier with a width of only about 200 meters, we used five RapidEye (5 m spatial resolution) and six PlanetScope (3 m) orthoimages acquired by Planet's constellation of CubeSats between 2009 and 2021. All the Planet scenes are ortho products and consist of visible and near-infrared frames. We also used the high-resolution Google Earth image to assist in the boundary delineation and detailed morphology characterization.

We used a stereo-pair acquired by the Hexagon KH-9 spy satellite to reconstruct the topography of KLP-37 in 1975. The KH-9 images (image IDs: DZB1211-500024L002001 and DZB1211-500024L003001), with a ground resolution of about 7.6 m, were scanned by the U.S. Geological Survey (USGS) at a resolution of 7 micrometers. The formation procedure of the KH-9 DEM using stereo-photogrammetry will be detailed in Section 3.2.1. We also orthorectified the KH-9 images, from which the glacier boundary was further identified.

In addition to the KH-9 DEM, we used another five DEMs in different periods between 2000 and 2018 to infer the elevation changes of the KLP-37 glacier after 1975 (see Table 1). Of the five DEMs, three were obtained from optical stereo-image photogrammetry, and the other two were generated from InSAR. The SRTM and HMA DEMs are publicly available, and the

commercial TanDEM-X DEM is provided by the German Aerospace Center. The latest ASTER DEM (2018) was generated from an ASTER stereo-image pair acquired by the Terra satellite (Hirano et al., 2003). Two 8-meter High Mountain Asia (HMA) DEMs (2010 and 2014) were generated from very-high-resolution imagery from Worldview-1/2 satellites by Shean (2017). The SAR interferometry derived DEMs include the SRTM (2000) and TanDEM-X DEMs, with spatial resolutions of 30 m and 12 m, respectively (Farr et al., 2007; Krieger et al., 2007). Two kinds of SRTM DEM data exist, which are generated from radar data with different wavelengths (i.e., C-band and X-band). Here both the C- and X-band SRTM DEMs were used to correct for the penetration depth of radar wavelength. The commercial TanDEM-X DEM was produced from the TerrSAR-X/TanDEM-X SAR images acquired between January 2011 and September 2014, representing an average estimate over the period. We, therefore, did not use the TanDEM-X DEM to calculate elevation changes of the glacier surface. Instead, the TanDEM-X DEM was taken as a reference to evaluate the accuracy of the other DEMs because of its relatively high vertical accuracy (~2 m) (Kumar et al., 2020; Riegler et al., 2015). The TanDEM-X DEM was also used to extract the elevations of the selected ground control points (GCPs) when constructing DEM with the ASTER stereo-images (see Section 3.2.1).

## 3.   Methodology

### 3.1 Derivation of glacier terminus changes

We manually digitized the boundary of the glacier from the orthorectified KH-9 image and the 11 Planet orthoimages. The boundaries of the snow-covered part of the KLP-37 glacier were cross-checked against the GAMDAM glacier inventory (Sakai, 2019). We identified the center point of the glacier terminus in each image and then estimated the advance distance between two consecutive periods. To avoid the influence of spatial resolution on the determination of the terminus center point, all the images were resampled into a common geometry with a spatial resolution of 3 m, the same as the resolution of the PlanetScope images. By dividing the advance distance by each image pair's time span, we further estimated the terminus advance velocities ($v_t$) during 1975 and 2019. The uncertainty of the velocity estimate can be written as

$$\varepsilon_{v_t} = (\sqrt{r_1{}^2 + r_2{}^2} + \varepsilon_{geo})/\Delta t \tag{1}$$

where $r_1$ and $r_2$ are spatial resolutions of the two images, respectively; $\varepsilon_{geo}$ is the relative georeferencing error between the two images (Hall et al., 2003; Rashid et al., 2020); $\Delta t$ is the time span of the two images. Note that we estimated $\varepsilon_{geo}$ for the Planet image pairs from off-glacier cross-correlations, and for the KH-9–Planet image pair from coordinate differences at the selected ground control points.

### 3.2 Surface elevation changes from DEM differencing

### 3.2.1 DEM extraction from stereo images

We used the HEXIMAP toolbox, developed by Maurer et al. (2016) and coded in MATLAB with an automated pipeline, to generate DEMs from the KH-9 stereo images. HEXIMAP combines computer-vision concepts with traditional

photogrammetric methods to achieve a satisfactory solution of DEM accuracy. The OpenCV library is used in HEXIMAP for surface feature matching, uncalibrated stereo rectification, and semiglobal block matching. Each digital KH-9 image provided by the USGS consists of two sub-frames with some overlap. The preprocessing steps thus include the stitching of sub-frames and cropping to the region of interest. Because the exterior parameters of KH-9 images are unavailable, HEXIMAP first generates a DEM with only rough geographical coordinates and then refines the DEM by matching it to an external reference DEM (Maurer et al., 2016). Here we used the TanDEM-X DEM as the reference and finally extracted the KH-9 DEM with a spatial resolution of 15 m. Fig. S1 (see the Supplementary file) shows the generated KH-9 DEM for the study area.

We used the open-source ASP (Ames Stereo Pipeline, v2.6.2) software developed by NASA to extract DEMs based on the ASTER stereo images. The ASP software provides a program called "aster2asp", which implements a straightforward pipeline for processing ASTER stereo images and extracting DEM (Shean et al., 2016). To ensure the DEM accuracy, we selected 12 GCPs on the high-resolution Planet image (2016/10/10) and determined the elevations of GCPs with the TanDEM-X DEM data. The ASTER DEM was also generated with a spatial resolution of 15 m.

### 3.2.2 DEM co-registration and differencing

With the KH-9 (1975), SRTM (2000), HMA (2010 and 2014), and ASTER (2018) DEMs, we formed four pairs with consecutive times to perform DEM differencing. All the DEMs were resampled into the overlap region shown in Fig. 1b (see the white box) to a spatial posting of 15 m. The DEM pairs need to be co-registered to minimize the errors associated with geometric shifts. We used the method that relies on the geometric relationship between the shift vectors and the slope and terrain aspect to coregister the DEM pairs (Nuth and Kääb, 2011). The glacierized regions and the area with a slope smaller than 10° were excluded before the coregistration. Fig. S2 shows an example of DEM differences before and after the co-registration with the two HMA DEMs (see Table 1), demonstrating a remarkable reduction of residuals due to geometric shifts between DEMs. After the co-registration, the older DEM of a pair was then resampled using the cubic interpolation method with a resampling posting of 15 m. Elevation differences were calculated by subtracting the older DEM from the younger DEM such that glacier thickening values are positive. We also calculated volume changes over the glacier tongue area (the yellow polygon in Fig. 2a) during different periods based on the elevation change estimates.

Two possible systematic biases resulted from (1) the penetration of radar waves and (2) the seasonal snow cover on the glacier should be corrected in DEM differencing. We used the SRTM-C DEM to calculate elevation changes between 1975/12/12–2000/02 and 2000/02–2010/12/05. Considering the low penetration depth of X-band radar into snow/ice surface, we used the X-band SRTM DEM (SRTM-X) as a reference to correct for the bias of the SRTM-C DEM. The elevation differences over the KLP-37 glacier between the X- and C-band SRTM DEMs were calculated, from which we found that the elevation difference increased with elevation (Fig. S3). Huge variations of the elevation differences can also be observed (Fig. S3), highlighting the heterogenous penetration differences between the two SRTM DEMs. The mean elevation differences were about 0.43 m and 2.36 m for regions below and above 5000 m, respectively. The small penetration depth in the glacier tongue region is

probably due to the debris cover on the surface. Following previous studies (e.g., Li et al., 2021), we implemented a linear fitting to the elevation differences and then applied the penetration correction with the fitted model (i.e., $y = 0.046h - 21.5335$).

Snow cover changes due to the different acquisition times of DEMs may also contribute to the estimate of glacier elevation change (Gardelle et al., 2013). This effect is usually referred to as the seasonality artifacts in DEM differencing. All the DEMs we used were acquired during winter except for the HMA DEM of late August in 2014. We did not apply adjustments to winter-winter DEM pairs because KLP-37 is a summer-accumulation type glacier. However, elevation changes for the summer-winter DEM pairs (i.e., HMA10-HMA14 and HMA14-ASTER) need to be corrected by considering four to five months of time differences. Due to the scarcity of snow accumulation documentation on the glacier, we conservatively adjusted the HMA DEM in 2014 by applying a bias correction of 0.1 m per month based on the remote-sensing derived seasonal snow depth in the KLP region (Tian et al., 2014).

### 3.2.3 Uncertainty assessment

Elevation change uncertainty estimates were calculated based on off-glacier elevation changes in the DEMs' overlap region (see the white rectangle in Fig. 1b). We calculated the uncertainty statistically by dividing the altitude into different bands with a 50 m interval. We assumed that the error for each pixel of elevation change ($\varepsilon_{\Delta h}^i$) is equal to the standard deviation of each elevation band, which can be calculated according to standard principles of error propagation (Gardelle et al., 2013)

$$\varepsilon_{\Delta h}^i = \frac{\sigma_{\Delta h}^i}{\sqrt{N_{\text{eff}}}} \ , \tag{2}$$

where $\sigma_{\Delta h}^i$ is the standard deviation of the elevation changes in the $i_{th}$ elevation band; $N_{\text{eff}}$ represents the number of independent values in the band, which can be calculated as

$$N_{\text{eff}} = \frac{N_{\text{tot}} \cdot P_s}{2 \cdot d} \ , \tag{3}$$

where $P_s$ is the pixel posting of the DEM; $N_{\text{tot}}$ is the total number of elevation change measurements in the elevation band; $d$ is the distance of spatial autocorrelation of the elevation change maps, which can be obtained by a least-square fit to the experimental, isotropic variogram of all off-glacier elevation differences (Wang and Kääb, 2015; Magnússon et al., 2016). The autocorrelation distances for the four DEM pairs were 286 m (KH-9–SRTM), 167 m (SRTM–HMA10), 189 m (HMA10-HMA14), and 909 m (HMA14-ASTER), respectively, with a mean value of about 388 m, similar to the typical value of about 500 m by previous studies (McNabb et al., 2019). The error of the glacier volume change ($\varepsilon_{\Delta V}$) was derived from the uncertainty of elevation change:

$$\varepsilon_{\Delta V} = \sqrt{\sum (A_i \cdot \sigma_{\Delta h}^i)^2} \ , \tag{4}$$

where $A_i$ is the area of each elevation band.

Page 9

### 3.3 Surface velocity from image cross-correlation

To investigate the dynamics of the KLP-37 glacier tongue, we applied cross-correlation to the orthorectified Planet images to obtain two-dimensional surface displacements. The acquisition 2015/10/12 and 2021/03/23 were not used because the glacier tongue was partially snow covered. Seven consecutive image pairs were formed and correlated for obtaining surface velocities during each period. We extracted the near-infrared band of the Planet images, i.e., the sub-band that has the longest wavelength, to implement the correlation measurement. This is because the long-wavelength band is generally less affected by cloud and has a higher radiometric sensitivity.

The freely available, open-source software Micmac was used to implement the sub-pixel image cross-correction (Rosu et al., 2015; Rupnik et al., 2017). The correlator program "MM2DPosSism" provided in Micmac employs a hierarchical matching scheme using normalized cross-correlation (NCC) with a non-linear cost function to find the most likely match for each pixel. The matching cost function is evaluated from the NCC coefficient considering only correlation coefficients $C \geq C_{min}$. Micmac also adopts a unique regularization parameter $r$ to smooth displacement field and reduce noise and outliers, which allows the use of smaller matching template windows targeting small landscape features. Here we set values of 0.5 and 0.3 for $C_{min}$ and $r$, respectively, and a moving window of 9×9 pixels in the correlation processing. In addition, we specified the main flow direction (i.e, NE15°) as the privileged direction of regularization. The flow velocity's uncertainties primarily result from the imprecise matching of the surface features on the glacier. Similar to previous studies (Rupnik et al., 2017), we inferred the uncertainty of flow velocity using the correlation estimate at a stable and plain surface below the glacier terminus (see the white rectangle in Fig. 2a).

### 3.4 Avalanche Runout Hazard Assessment

We estimated the maximum runout distance to quantitively assess the possible influence of the glacier detachment. We first estimated the maximum runout distance using the angle of reach (also called "Fahrböschung"), which is defined as arctan ($H/L$), whereas $L$ is horizontal reach of avalanche mass and $H$ is elevation drop. Note $H$ is measured from the avalanche start point to the stop point. Previous investigations have shown that Fahrböschung value for low-angle glacier detachments typically ranges between 5° and 10° (Kääb et al., 2021a). Given the possible avalanche start and stop points, we can thus roughly estimate the maximum runout distance.

We also quantitively estimated the extent of hazard-prone areas using avalanche-dynamics modeling. We employed the Voellmy-Salm model to simulate the possible runout extent and flow height of ice materials. The Voellmy-Salm model was originally developed to investigate the detailed flow patterns and dynamics related to pure snow valances (Bartelt et al., 1999); while the model has also been widely used to simulate the runout distance of glacier/ice avalanche events (Allen et al., 2009; Bai and He, 2020; Evans et al., 2009). The Voellmy-Salm model divides avalanche flow resistance into a speed-independent Coulomb-type friction (friction coefficient $\mu$) and a velocity-dependent, turbulent friction (friction coefficient $\xi$, units: m·s$^{-2}$)

(Bartelt et al., 1999). Here, we determined the ranges of $\mu$ and $\xi$ from previous studies because it is impossible to obtain these friction parameters directly. Retrospective analyses of glacier/ice avalanche events from different glacial environments based on the Voellmy-Salm model have shown that the best-fit frictional values generally range between 0.05 and 0.2 for $\mu$, and between 1000 and 4000 m·s$^{-2}$ for $\xi$ (Allen et al., 2009). Studies of a few glacier detachment events also have revealed friction parameters laying within the above ranges. For instance, the best-fit $\mu$ for the Aru-1 and Aru-2 glacier detachments are 0.11 and 0.14, respectively (Kääb et al., 2018); the 2002 Kolka detachment has best-fit values of 0.05 and 2700 m·s$^{-2}$ for $\mu$ and $\xi$, respectively (Allen et al. 2009).

We used the open-source software MASSFLOW to implement the modeling (Ouyang et al., 2013). MASSFLOW contains the Voellmy-Salm model and allows for the simulation of rapid mass movements accounting for momentum and including processes of friction, fluidization, and erosion. The input datasets of the Voellmy-Salm model consist of a DEM for generating a meshed grid and a source file containing the geographical extent and thickness of the avalanche materials. Here we used the 7-meter HMA DEM acquired in 2014 as the input topography. The ice thickness was derived from Farinotti et al. (2019), who provided an ensemble-based estimate for the ice thickness distribution of all glaciers included in the Randolph Glacier Inventory (RGI) apart from the Greenland and Antarctic ice sheets. However, the ice thickness for KLP-37 does not include the tongue region. Given that the ice thickness in KLP-37 shows a homogenous pattern, with 86% of pixels having a thickness ranging between 25 and 45 m, we took the mean value of the known pixels (i.e., 32±6 m) as the glacier tongue ice thickness.

## 4. Results

### 4.1 Morphological changes and terminus advance

The optical KH-9 and Planet satellite images enable us to inspect the morphological changes of the KLP-37 glacier in the past 46 years. A time-lapse of the optical images covering the full glacier are shown in Fig. S4, and the zooms of the glacier tongue region for a clear inspection are shown in Fig. 3. Satellite images show that the transverse crevasses in the glacier cirque had commenced in 1975 and became more evident in the following years (see the red arrows in Fig. S4). Specifically, the crevasses' length and width remarkably increased after 2013. The presence of crevasses in the glacier accumulation region since the 1970s and the widening of the crevasses in recent years indicate that the glacier may develop into a less stable regime. Also, the glacier tongue exhibits a swollen body with a steep front in the KH-9 image (Fig. 3a), indicating a large amount of ice mass had been deposited there by 1975. However, the swollen body has become falt in 2021 (Fig. 3n), implying the loss of ice in the past decades and that the glacier is in a negative mass balance state.

Satellite optical images also clearly show the evolution of the ice-dammed lake (the light blue polygons in Fig. 3) developing in the front of the glacier's west branch. The lake was not visible in the 1975 image. However, we cannot rule out the possibility that the coarse resolution (~ 7.6 m) of the KH-9 image may hinder the identification of the lake. The high-resolution Google Earth image in 2005 shows that the ice-dammed lake had appeared before then (Fig. S5). To investigate the changes in lake

area over time, we delineated the boundary of the lake based on the Planet and Google Earth images and estimated the lake area between 2005 and 2019 (see Table S1). The uncertainty in delineating the lake was obtained from five independent digitizations (Paul et al., 2017). The lake area in summer was commonly larger than that in winter, and the area larger than 10000 m$^2$ all occurred during July and September. Also, the lake area showed an expansion trend in recent years, with the smallest value (3745±229 m$^2$) in the winter of 2010 and the largest value (21276±1646 m$^2$) in the summer of 2017. A small supraglacial pond developed in the depressions on the glacier tongue surface; The pond (Figs. 2a and 3) appeared in 2013 and disappeared in 2017.

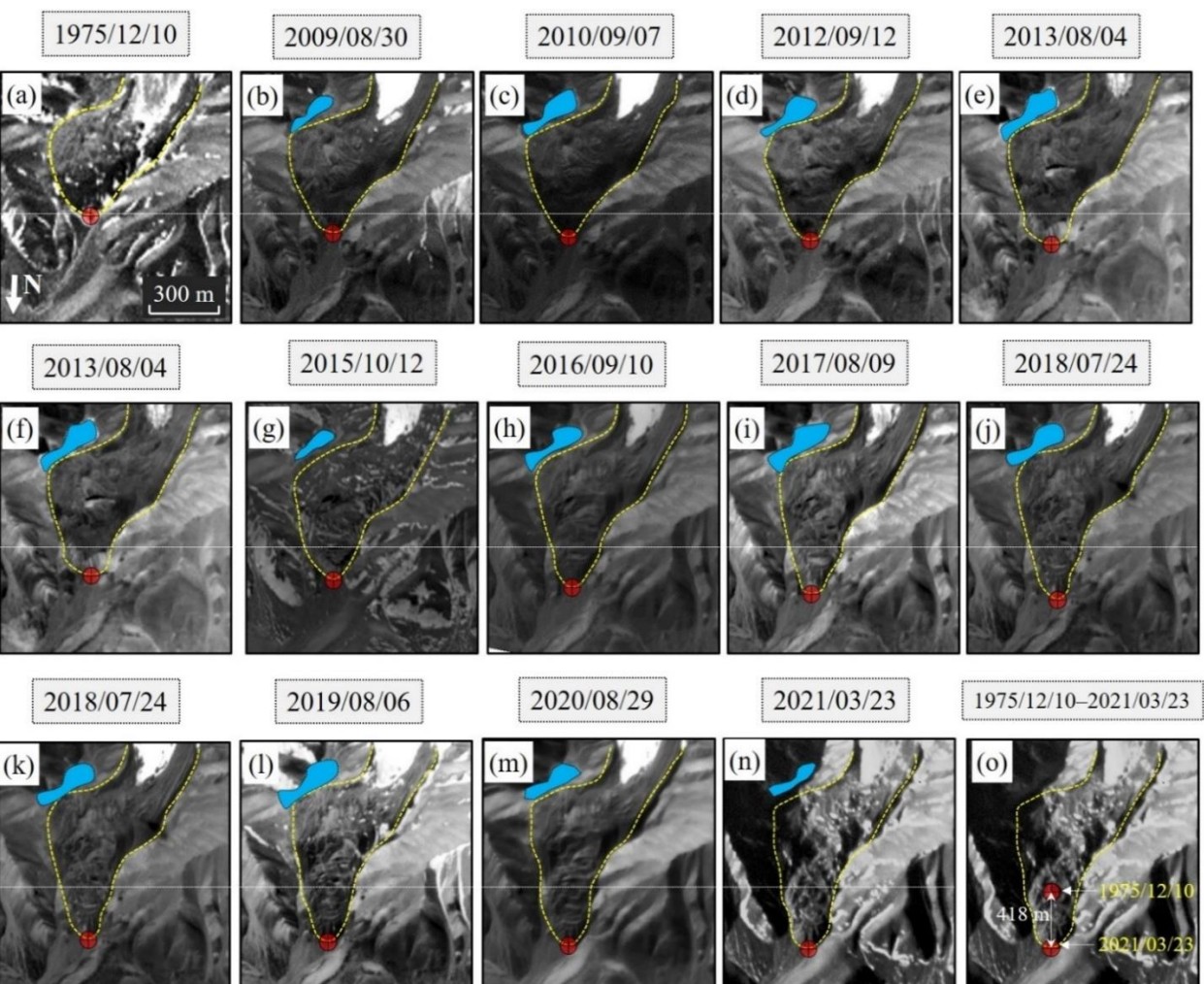

**Figure 3: Optical satellite images covering the KLP-37 glacier tongue. (a) Hexagon KH-9 image acquired on 1975/12/10 (© U.S. Geological Survey); (b-n) Planet images acquired between 2009/09/07 and 2021/03/23 (© Planet Labs); (o) Comparison of the glacier terminus position between 1975 and 2021. The horizontal white dotted lines provide a reference for the terminus position in 1975. The blue polygons represent the ice-dammed glacier lake.**

The glacier terminus showed a stepwise advance (see the red points at the glacier snout in Fig. 3) between 1975 and 2021. Table S2 lists the terminus point coordinates in each satellite image, and Fig. 4 shows the changes in terminus advance velocities during periods between the consecutive image acquisitions. The glacier tongue moved downstream and narrowed due to the topographic blocking on both sides. The glacier terminus' total advance distance was about 418±24.13 m in the past 46 years (Fig. 3o). The advance velocity between 1975 and 2009 was about 4.55±0.46 m·a$^{-1}$, and the velocity rose to more than 10 m·a$^{-1}$ after 2009. The velocities were stable within each period of 2009–2015 and 2015–2021 but jumped between 2015 and 2016. The mean velocity was 15.70±3.02 m·a$^{-1}$ during 2009–2015 and 30.88±2.36 m·a$^{-1}$ during 2015–2021.

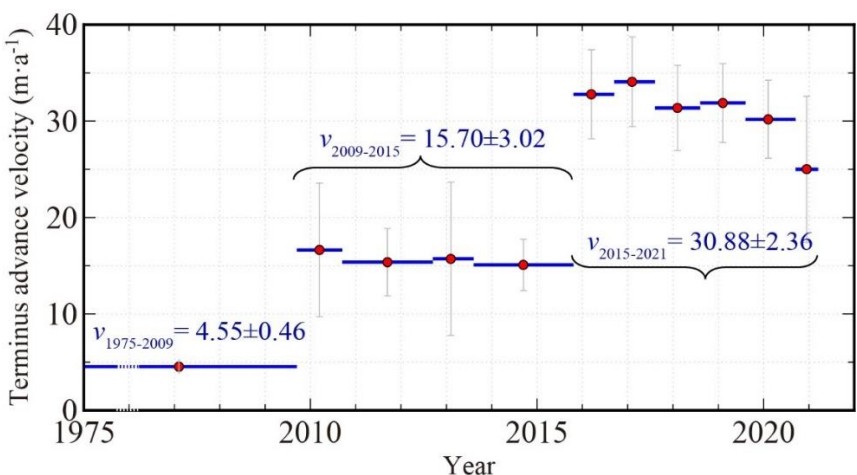

**Figure 4: Terminus advance velocities and the associated uncertainties (the light gray error bars) of the KLP-37 glacier estimated from KH-9 and Planet images. We also annotated the mean velocities during three periods: 1975–2009, 2009–2015, and 2015–2021. Note that the time axis during 1975–2009 is not equally posted for a better presentation.**

### 4.2 Surface elevation and volume changes

We evaluated the accuracies of the DEMs before DEM differencing using the TanDEM-X DEM as a reference. Most of the elevation differences range between -10 and 10 m (see Fig. S6). The mean values of the off-glacier elevation differences between the TanDEM-X DEM and other DEMs are all smaller than 0.2 m except for the ASTER DEM (-0.31 m), indicating that the accuracies of the DEMs we used are feasible for inferring the elevation changes of the KLP-37 glacier by DEM differencing. The large errors in the ASTER DEM are mainly distributed near the mountain ridges and steep slopes, which thus would not significantly affect the elevation differences over the KLP-37 glacier.

Surface elevation changes of the KLP-37 glacier from DEM differencing overall exhibit thinning in the glacier source region and thickening in the glacier tongue (Figs. 5a-d). The void regions mainly appear in the accumulation area of the glacier's west branch, where the slopes are steep and intense crevasses developed. We selected a specific point ′T′ with a window size of 3×3 pixels (~ 2000 m$^2$) in the center of the accumulation region (see Fig. 5a) and found that the elevation differences at this

point for the KH-9–SRTM, SRTM–HMA10, HMA10–HMA14, and HMA14–ASTER pairs are -1.64±0.77, -0.22±0.15, -0.90±0.31, and -0.02±0.34 m·a$^{-1}$, respectively. Although the estimate from the HMA14-ASTER DEM pair had considerable uncertainty, the elevation change rate at point ″T′″ showed a decreasing trend during 1975–2018. The elevation differences at the point ″T′″ between the X-band and C-band SRTM is 2.44 m. Considering that we had applied a correction of 2.13 m (given the correction model of $y=0.0046×h-21.5335$ and the elevation of 5145 m at the point ″T′″) to the SRTM DEM, a residual of 0.31 m would remain in the elevation change estimates. We thus inferred that the elevation change errors due to SRTM penetration were about 0.01 m·a$^{-1}$ and 0.03 m·a$^{-1}$ for the KH-9–SRTM and SRTM–HMA10 DEM pairs, respectively.

The east branch of KLP-37 did not retreat in the past 40 years, but elevation differences there show an overall thinning pattern. The mean elevation changes over the whole east branch were about -0.18±0.12, -0.32±0.15, -0.48±0.09, and -0.82±0.55 m·a$^{-1}$ for the four DEM pairs, respectively. The continued thinning of the glacier's east branch presumably explains the expansion of the ice-dammed glacier lake, which develops in the front of the east branch and receives the meltwater from the glacier directly.

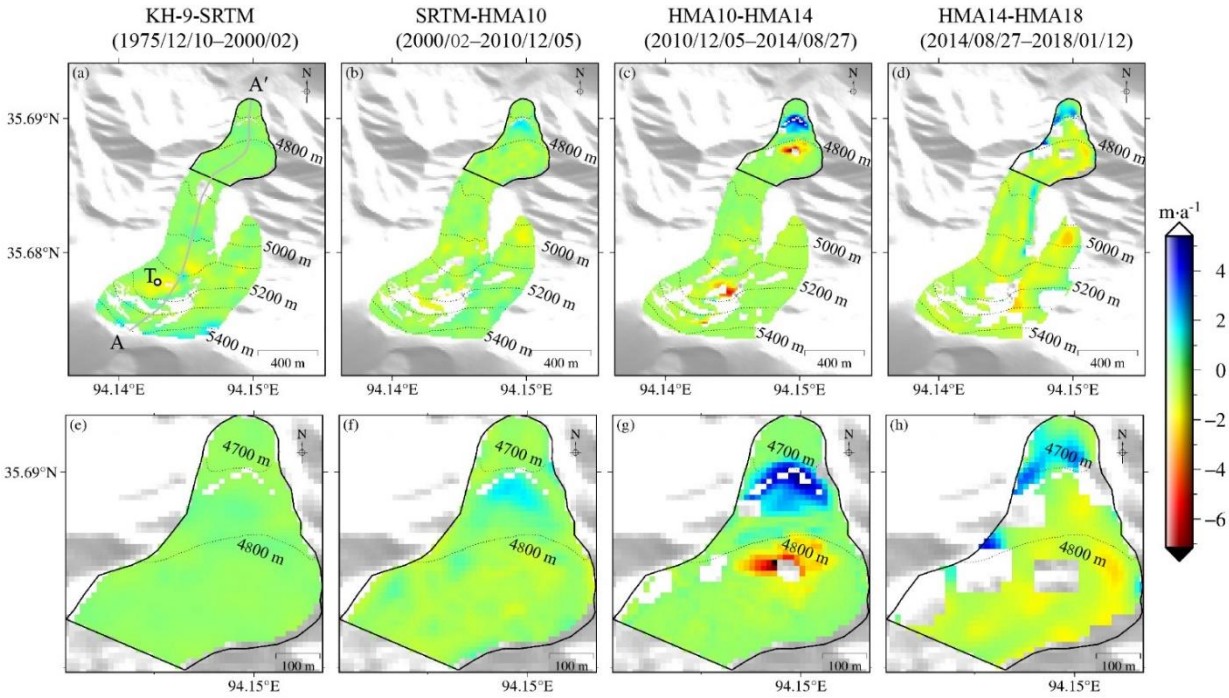

**Figure 5: Surface elevation changes of KLP-37 glacier between 1975 and 2018. (a-d) Elevation changes over the full glacier for the DEM pair KH-9-SRTM (1975–2000), SRTM-HMA10 (2000–2010), HMA10-HMA14 (2010–2014), and HMA14-ASTER (2014–2018), respectively. (e-f) Amplified elevation changes over the glacier tongue region for the four DEM pairs. The black dotted lines indicate the elevation contours.**

The elevation changes over the west-branch glacier tongue region are enlarged and shown in Figs. 5e-h. We can observe elevation increases in the front part of the tongue and widespread thinning just above the elevation thickening area. Specifically, thinning was prevalent between the 4750 m and 4880 m for the SRTM–HMA10 and HMA14–ASTER DEM pairs. The snout

of the elevation thickening area continually advanced from 1975 to 2018, with a remarkable advance during 2014–2018. Elevation changes confirm the stepwise glacial terminus advance (see Section 4.1) and imply a slow surge-like mass transfer process in the tongue area. This kind of flow behavior resembles the surging movement of the surge-type glacier, which has also been reported on the Aru or Amney Machen glaciers preceding the occurrences of ice detachments (Kääb et al., 2021a; Paul, 2019). However, we cannot conclude that KLP-37 is a surge-type glacier because we did not capture periodical

alternations between long periods of slow flow and short periods of fast flow. We calculated the volume changes over the glacier tongue and found that the volume decreases were higher than the increases for all the four DEM pairs (Table 2). The net volume changes calculated from the four DEM pairs were -1.54±1.18, -5.74±1.47, -0.49±0.11, and -3.44±1.87 ×10$^5$ m$^3$, respectively, with a total net volume change of -11.21±2.66 ×10$^5$ m$^3$, indicating the continued loss of mass over the glacier tongue region.

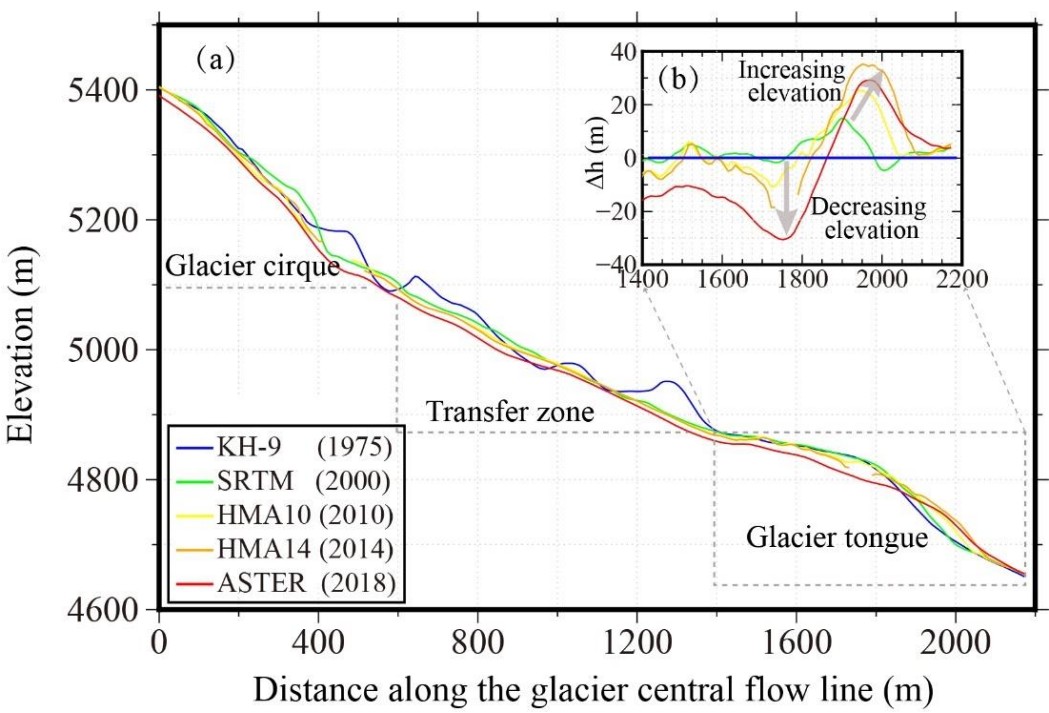

**Figure 6: Glacier surface topography extracted from the five DEMs along the glacier central flow line AA′ (Fig. 5a). The inset shows the elevation changes along the profile with respect to the KH-9 DEM in 1975.**

Fig. 6 shows the changes of surface topography from 1975 to 2018 along the glacier's central flow line (see AA′ in Fig. 6a). We divided the glacier into three parts: glacier cirque (> 5100 m), transfer zone (4880–5100 m), and the glacier tongue (4700–

4880 m). The cirque zone has a mean steep slope angle of about 34°, while the slope becomes gentle in the transfer zone with a mean slope of 19°. Both the glacier cirque and transfer zone exhibited an overall subsidence pattern. The glacier tongue can be further divided into thinning and thickening parts at an elevation of about 4800 m. Notably, the thinning rate in the glacier tongue region was much higher than that in the transfer zone (see the subsidence arrow in Fig. 6b). The part above 4800 m continually subsided while the lower part showed increasing elevation, with the maximum of elevation changes moving toward the glacier terminus progressively (Fig. 6b).

**Table 2. Net volume changes over the glacier tongue region calculated from the four DEM pairs.**

| DEM pairs | Time span (years) | Net volume change ($\times 10^5$ m³) |
|---|---|---|
| KH-9–SRTM | 24.2 | -1.54±1.18 |
| SRTM–HMA10 | 10.8 | -5.74±1.47 |
| HMA10–HMA14 | 3.73 | -0.49±0.11 |
| HMA14–ASTER | 3.38 | -3.44±1.87 |

Uncertainties associated with the glacier surface elevation changes were evaluated with the statistics of off-glacier elevation differences based on Equation (2). The uncertainties of elevation changes generally increase with elevation (Fig. 7). The mean uncertainties of elevation changes over the glacier tongue region (4700–4880 m) were about 0.04, 0.08, 0.02, and 0.41 m·a⁻¹ for the KH-9-SRTM, SRTM-HMA10, HMA10-HMA14, and HMA14-ASTER DEM pairs, respectively. Elevation changes in the DEM accumulation region (>5100 m) had relatively higher uncertainties, with mean values of 0.06, 0.06, 0.03, and 0.54 m·a⁻¹ for the four DEM pairs.

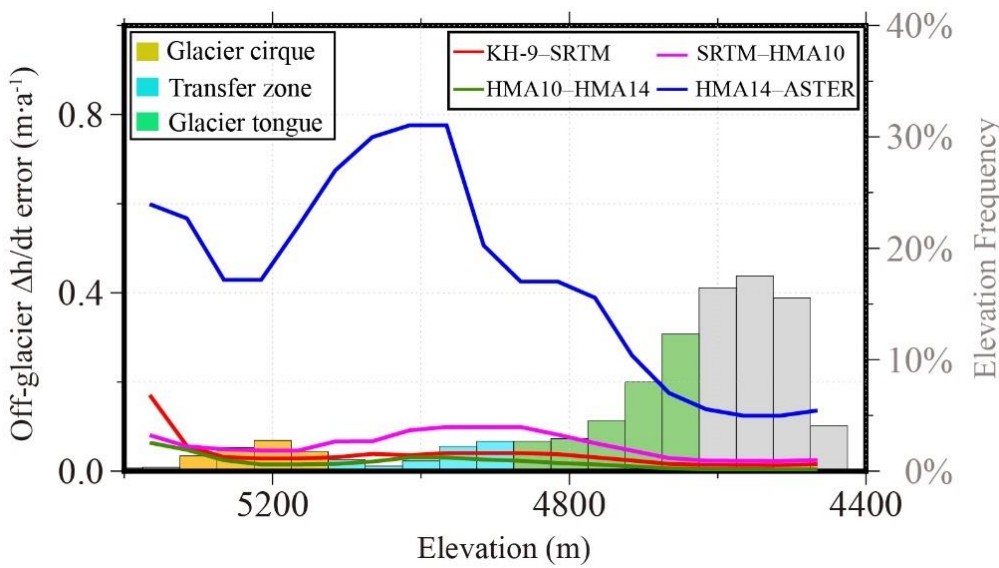

**Figure 7: Uncertainty of elevation change estimate as a function of elevation bands (the lines) in the off-glacier region. The right axis annotates the frequency number of each elevation band. The elevation bands corresponding to the glacier cirque, transfer zone, and glacier tongue ranges (see Fig. 6) are filled with yellow, cyan, and green colors, respectively.**

## 4.3 Surface flow velocities

Fig. 8 shows the surface flow velocities of the KLP-37 glacier tongue from the cross-correlation of seven Planet image pairs. We did not show the correlation result for the image pair 2013/08/04–2016/09/10 because 56% of the pixels within the tongue region have null values due to the low correlation coefficient, probably resulting from the remarkable change of surface features on the glacier. The mean uncertainty of flow velocity inferred from image correlations at the reference region (see Fig. 2a) was about 2.4 m. The glacier accelerated during 2009/08/30–2020/08/29 (Fig. 8), which is consistent with the advance pattern of the glacier terminus (see Section 4.1). The flow of the glacier was distinguishable from the surrounding area for velocity field after 2016, especially at the glacier front region. The maximum flow velocity within the glacier tongue area reached about 30 m·a$^{-1}$ after 2016, comparable to the estimated snout advance rate (30.88±4.45 m·a$^{-1}$) between 2015 and 2021.

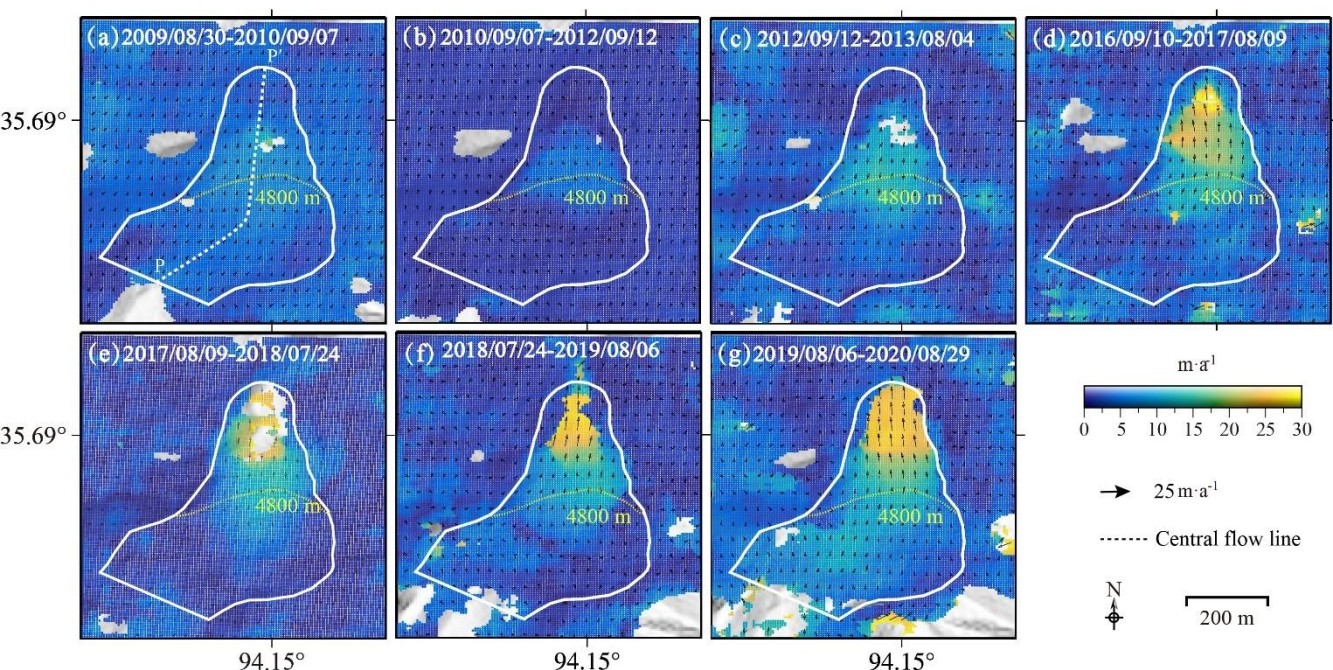

**Figure 8: Surface velocities over the glacier tongue of KLP-37 from image cross-correlation based on the seven Planet image pairs. The profile PP′ represents the central flow line, and the arrows mark the flow direction.**

We found that the velocity field showed different spatio-temporal patterns below and above 4800 m, where the glacier flow direction changed. Peak velocity in each observation period was observed in the lower part of the glacier tongue. Evident flow acceleration occurred below 4800 m between 2013 and 2016. The maximum velocities within the glacier tongue (the white

polygon in Fig. 8) were about 15.3±2.1 m·a⁻¹ (2012/09/12–2013/08/04) and 29.4±3.2 m·a⁻¹ (2017/08/09-2018/07/24) for the periods before and after 2013, respectively. Note the maximum values were determined on a pixel-by-pixel basis. Fig. 9 shows the velocity and topography variations along the central profile PP′ (location shown in Fig. 8a). The velocities on the profile were relatively stable above 4800 m before 2019 with a mean velocity of 4.6±1.5 m·a⁻¹, while the mean value velocity doubled (9.8±1.4 m·a⁻¹) during 2019–2020. The mean velocities below 4800 m for the periods of 2009–2010 and 2019–2020 were 6.3±1.8 and 22.3±3.2 m·a⁻¹, respectively. The more than tripled mean velocity below 4800 m in the past decade suggests that the glacier tongue may develop into a less stable regime.

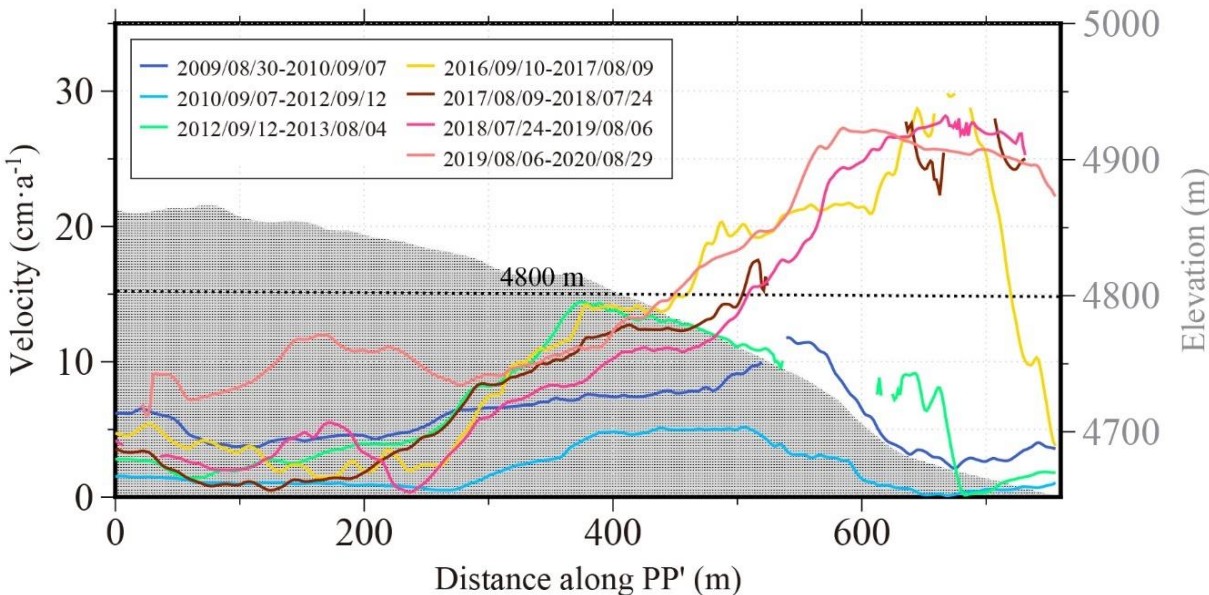

Figure 9: Surface velocities along the glacier central flow line PP′ (see Fig. 8a) during the seven image correlation periods. The light-blue-shaded area indicates the surface topography (right axis) along the profile.

### 4.4 Hazard assessment for the glacier detachment

The KLP-37 glacier shares several similarities with the Aru-1 and Aru-2 Glacier. First, both exhibited continuous thinning in the source region and thickening in the tongue region. Second, the ice flow direction along all three glaciers changes due to the local topography. Also, the glaciers' widths gradually narrow from the source region to the tongue, thus resulting in a large amount of ice mass accumulating at the glacier front and further leading to large gravitational potential energy there. We here assessed the runout hazard of two endmember avalanche scenarios: (1) an avalanche starting from the crevasses in the accumulation region of KLP-37 by assuming that the whole glacier detaches and (2) avalanche of the glacier tongue where apparent flow acceleration was observed. Fig. 10a shows the ice thickness map of the two scenarios. By integrating the ice thickness over the whole glacier and the tongue area (Section 3.4), the avalanche volumes for the two scenarios were estimated

to be about $27.06\times10^6$ and $6.63\times10^6$ m³, respectively. We kept two decimals for these volume values since the increase of decimals would not remarkably affect the runout distance estimates.

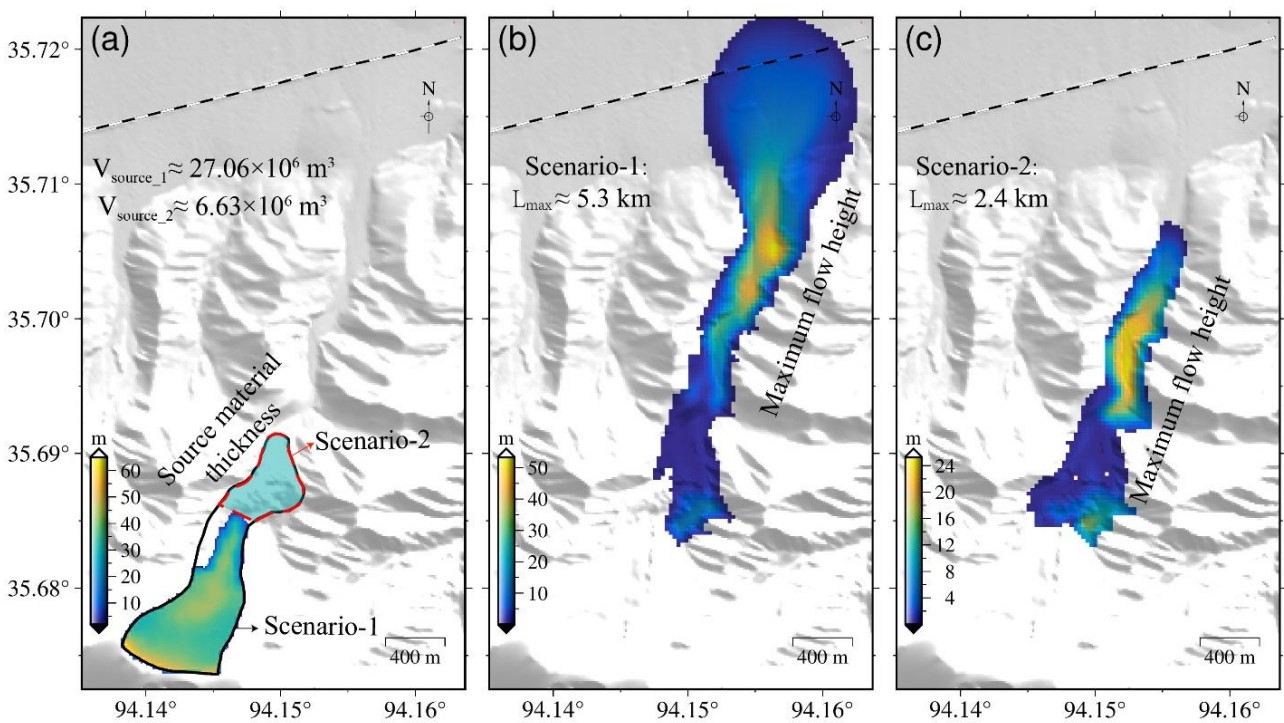

**Figure 10: Runout estimates for the detachments of the whole glacier (scenario-1) and glacier tongue (scenario-2). (a) Ice (i.e., source material) thickness distribution of the avalanche scenarios-1 (black polygon) and scenario-2 (red polygon) derived from the global glacier ice thickness products of Farinotti et al. (2019). (b) The simulated maximum flow height of the avalanche for scenario-1 when choosing moderate friction values of 0.15 for Coulomb friction (μ) and 2500 m·s⁻² for turbulent friction (ξ) in the Voellmy-Salm model. (c) The simulated results for scenario-2.**

Given that the angle of reach (i.e., Fahrböschung) for low-angle glacier detachments mainly ranges between 5° and 10°, we roughly estimated the lower and upper bounds of possible runout distance for the two detachment scenarios. We assumed dropping elevations of about 950 m (5350-4400 m) and 500 m (4900-4400 m) for the two avalanche scenarios, respectively (see Fig. 10a). The estimated maximum runout distance is thus about 5.4~10.9 km for detachment of the whole glacier, and 2.8~5.7 km for the avalanche of the glacier tongue region. Considering that the Qinghai-Tibet railway is about 5.0 km from the scarp in the glacier source region, we can infer that the detachment of the whole glacier (i.e., scenario-1) will easily reach the railway. However, whether the detachment of the glacier tongue would influence the railway ($L>4$ km) depends on the value of Fahrböschung angle (i.e., arctan($H/L$)), and a hazardous influence is only anticipated when the Fahrböschung angle is smaller than 7°.

Figures 10b and 10c show the maximum accumulation flow height of the two avalanche models with moderate values of 0.15 and 2500 m·s⁻² for $\mu$ and $\xi$, respectively. We found that the maximum flow heights are about 54 and 25 m for the avalanche scenario-1 and scenario-2, respectively. The maximum runout distances of the two scenarios are about 5.3 and 2.4 km (Figs. 10b and 10c), which are smaller than these estimated from the angle of reach. The Fahrböschung values for the two avalanche scenarios were calculated to be both about 9°, laying in the range of 5°~10° that was found on most detached glaciers. Similar to the assessment from the angle of reach, the Voellmy-Salm modeling results also show that the avalanche whole glacier can easily threaten the railway. For the avalanche scenario-2, the ice material would not affect the railway due to the small avalanche volume. However, considering that we cannot accurately determine the friction parameters, we will discuss how the variations of friction parameters would influence the runout distance estimate in Section 5.3.

## 5. Discussion

### 5.1 Classification of the landform in the glacier front

The "glacier tongue" we investigated here shows several features raising concerns about whether it is not part of the glacier but rather a landform in permafrost environments such as an ice-cored moraine or rock glacier. First, field investigations using ground-penetrating radar have shown that the lower permafrost altitude limit of the study region is about 4300 m, below the landform terminus (Wu et al., 2005). Global permafrost extent mapping also classifies the glacier font area as permafrost with a very high likelihood (Obu et al., 2019). Second, the landform shows a swallow body with a steep front, different from typical glacier tongues with gentle surface slopes. In addition, the surface debris cover makes it hard to discern whether the landform is connected with the upper glacier tongue. Recent studies have documented accelerations of rock glaciers and slope failures at rock glacier fronts in the French and Italian Alps (Eriksen et al., 2018; Kofler et al., 2021; Marcer et al., 2020). However, we here argue that the landform is part of the KLP-37 glacier tongue from both geomorphic and kinematic analysis below.

We first excluded the landform from the type of ice-cored moraine. A typical ice-cored moraine should be disconnected from the active glacier ice margin (Lukas, 2011). Regarding the landform at the front of KLP-37, however, the long-term advance of the glacier front without cutoff indicates that the inner body is connected with the glacier.

The KLP-37 glacier tongue also differs from a rock glacier. First, debris cover on a rock glacier is usually coarse, thick (>3 m). The field photo in 2016 (Fig. 2c) showed that the debris cover on the KLP-37 glacier tongue was thin and uniform. Second, we did not observe many ridges and furrows, the distinctive characteristics of rock glaciers, on the surface of the KLP-37 glacier front. In addition, rock glaciers typically move downslope at velocities smaller than 10 m·a⁻¹ (Kääb et al., 2021b; Wang et al., 2017), while the flow velocity of the landform from our cross-correlation measurement reaches ~30 m·a⁻¹.

### 5.2 Mechanisms of the glacier dynamics

Multi-temporal satellite imagery revealed that the glacier tongue of KLP-37 advanced progressively between 1975 and 2021,

and the snout advance velocity accelerated after 2015. From the analysis of changes in surface elevation and flow velocities, we suggest that the KLP-37 glacier tongue was undergoing a slow surge-like process during the observation period. We did not observe this process on the adjacent glaciers, indicating that the KLP-37 glacier has a unique glaciation setting causing the glacier tongue area to be highly active. Previous studies have shown that the factors resulting in glacier acceleration and even detachment mainly include the hydrothermal conditions of the glacier, topography, and the climate (precipitation/temperature) changes (Jacquemart et al., 2020; Kääb et al., 2021a; Kääb et al., 2018; Leinss et al., 2019). Next, we will discuss the possible mechanisms accounting for the dynamics of the KLP-37 glacier tongue.

The thermal regime of a glacier fundamentally influences its dynamics (Faillettaz et al., 2015; Pralong and Funk, 2004). We cannot determine whether the ice/bed interface is temperate because no temperature measurements beneath the KLP-37 glacier are available. However, the long preservation of the ice-dammed lake aside from the glacier tongue suggests that the glacier front is probably frozen to the underlying bedrock. This is supported by field investigations and permafrost mapping showing that the glacier tongue lays in a permafrost environment (see Section 5.1). The cold thermal regime for parts of glacier and fronts has also been found at detached glaciers such as the Aru twin glaciers, Leñas glacier, and Flat Creek glacier (Falaschi et al., 2019; Kääb et al., 2018; Jacquemart et al., 2020). The frozen base creates a favorable environment for ice accumulation and stress build-up, although it also increases the basal friction which is the threshold to be overcome for detachment occurrence. Detachment or acceleration of the glacier tongue occurs when the force balance cannot be achieved due to the increasing driving stress. The cross-correlations of the 2019–2020 image pair show that the mean velocities along the central profile were about 9.8±1.4 and 22.3±3.2 m·a$^{-1}$ for regions above and below 4800 m, corresponding to flow velocities of 2.7±0.4 and 6.1±0.9 cm·d$^{-1}$, respectively. These values are on the same order of magnitude as the estimated velocity of 2 cm·d$^{-1}$ for pure ice deformation (Leinss et al., 2019; Round et al., 2017). We thus suggest that the internal ice creeping should mainly account for the downslope movement of KLP-37. However, given the fast and homogenous flow in the lower part of the glacier tongue, it is reasonable to postulate that basal sliding is at least partly responsible for the flow dynamics.

The KLP-37 glacier's geometry presumably plays an important role in accounting for the slow surge-like behavior of the glacier. From the high-resolution optical images (Fig. 3), we can observe that the flow direction of the glacier changes from NE24º to NE5º at an elevation of about 4800 m due to the local topography. The glacier tongue downslope of this turning exhibits a "V" shape (the upper part is wide, while the lower part is narrow). The narrowing at the tongue provides a buttressing effect on the upstream ice mass. However, on the other hand, the specific shape of the glacier tongue prevents the glacier from adjusting its geometry to the changed driving stresses with the accumulation of ice mass. This thus continuously increases the stresses on the frozen terminus and margins, until reaching a critical point where the resisting force is eventually overcome and an acceleration or detachment occurs. The abrupt velocity increase below 4800 m between 2015 and 2016 could likely result from the mass accumulation, as evidenced by the apparent elevation increase above 4700 m from the DEM difference between 2010 and 2014 (Fig. 5g). Also, the mean slope angle is about ~10° between 4800 and 4880 m but increases to ~20° at the lower place (4700–4800 m), making the lower part favorable for glacier acceleration. Compared with the glaciers nearby,

the local topography at the KLP-37 front thus provides a preconditioning factor for the destabilization of the glacier tongue.

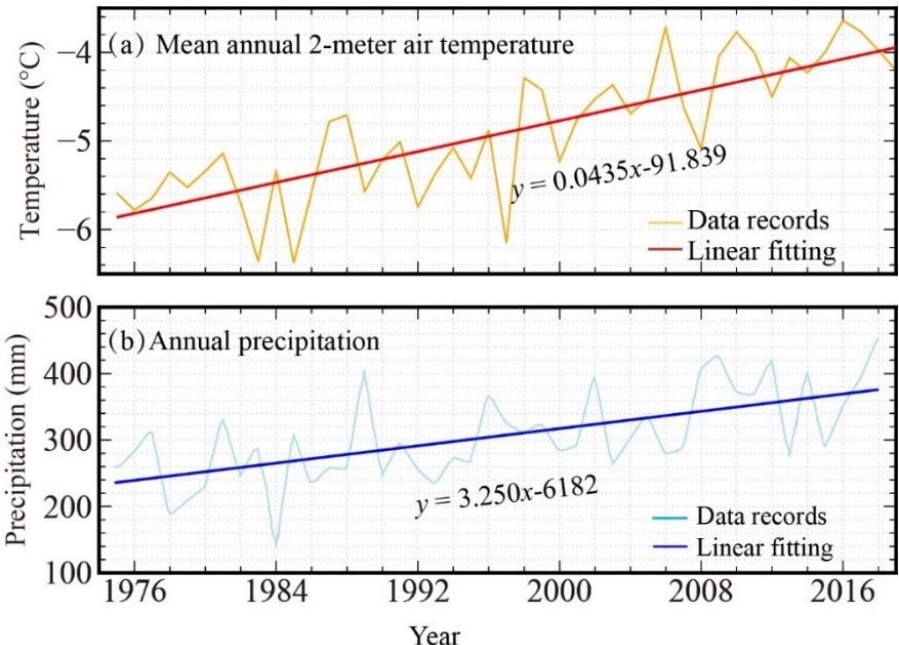

**Figure 11: Mean annual 2-meter air temperature (a) and annual precipitation (b) at the Wudaoliang meteorological station (4613 m above sea level) about 60 km south of the KLP-37 glacier. The equations annotated represent the best linear fitting model of the data records.**

The trend of a warmer and wetter climate in the past decades on the QTP may be the long-term driving factor for the continuous advance of the glacier tongue of KLP-37. Fig. 11 shows the mean annual 2-meter air temperature and mean annual precipitation at the Wudaoliang meteorological station about 60 km west (35.3°N, 93.6°E) to the glacier between 1975 and 2018. Both the air temperature and precipitation records show increasing trends: the increase rate of mean annual air temperature is 0.0435 °C·a$^{-1}$, while the precipitation has an increasing rate of about 3.250 mm·a$^{-1}$. Given the stepwise increase of snout advance velocity between 1975 and 2021 and the more than tripled mean flow velocity below 4800 m between 2009 and 2020 (Section 4), we suggest that the climate warming in the study region likely contributes to the long-term acceleration of the glacier tongue.

The expansion of the ice-dammed lake in the past decade (Section 4.1) also justifies the study area's warming and wetting trend. The seasonal fluctuations of the lake area, the disappearance of the supraglacial pond, and the presence of crevasses and ice cliffs in the glacier front imply that the water drainage also likely affects the glacier dynamics. Lake water may penetrate through the crevasses into the glacier bed, leading to the increase of basal water pressure and further weakening the subglacial till (Jacquemart et al., 2020). In addition, liquid water embedded in the glacier can lead to a decrease in basal friction (Kääb et al., 2021a). However, verifying such hydrological effects is difficult and needs further dense observations on the surface

movements and water availability.

Previous studies have revealed that climate warming and increased rainfall can promote glacier movement and eventually lead to glacier detachments (Bai and He, 2020; Kääb et al., 2018; Tian et al., 2016). Taking the Aru-1 glacier as an example, the regional climate warming was likely the reason for changing the glacier from retreat to slow advance in 2013 (a total advance of about 300 m before the ice avalanche in 2016) (Tian et al., 2016). Specifically, heavy precipitation accounting for 90% of the total precipitation of 2016 was recorded during the 40 days prior to the Aru-1 glacier detachment, and the extreme precipitation was suggested to be the triggering factor for the detachment (Tian et al., 2016). The increase of meltwater in summer caused by climate warming could increase the overload of glacier surface and the supply of liquid water into the sliding surface, thus further promoting the downward movement of the glacier (Leinss et al., 2019).

In summary, we suggest that the cold glacier front, the particular local topography, and the long-term climate change in the East Kunlun Mountains are the main factors controlling the dynamics of the KLP-37 glacier tongue. With the warming and wetting trend of the regional climate, the risk of detachment of the KLP-37 glacier tongue may threaten the safety of the nearby Qinghai-Tibet railway and highway.

## 5.3 Limitations and implications for glacier detachment hazard assessment

Due to the limited data on ruout modeling for low-angle glacier detachments, we ran the Voellmy-Salm model by specifying moderate friction parameters determined from the previous modeling of ice/rock avalanche events. Our modeling results show that the avalanche of the whole glacier would impact the Qinghai-Tibet railway, similar to the hazard assessment based on the angle of reach. The modeling results also show that the avalanche of the glacier tongue would not reach the railway. However, it should be noted that the runout distance from Voellmy-Salm modeling depends on the selection of friction parameters. To investigate how the altered frictional input parameters would influence the runout distance estimate of the avalanche scenario-2, we ran the Voellmy-Salm model with multiple combinations of the parameter values. The result depicted in Fig. 12 shows that the runout distance would be longer with decreasing $\mu$ and increasing $\xi$. Only when $\mu$ is lower than 0.1, the avalanche of the KLP-37 glacier tongue would pose a threat to the Qinghai-Tibet railway (with a runout distance longer than 4 km). Note that our modeling did not include the lubrication effects of fine-grained sediments under the glacier, which may reduce the avalanche friction and allow the detachments to accelerate particularly fast and cover long distances (Kääb et al., 2021a). It is thus essential to monitor the dynamics of the KLP-37 glacier continually in combination with numerical simulation to predict its potential hazardous impacts.

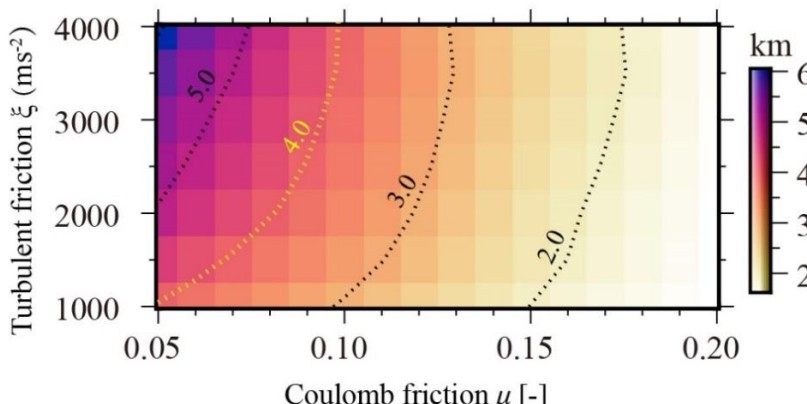

**Figure 12: Variations of runout distance for varying friction parameters ($\mu$ and $\xi$) of the Voellmy-Salm model. The dotted lines represent the contour lines with the yellow one representing the runout distance (4 km) that will threaten the Qinghai-Tibet railway.**

Although most of glaciers in QTP have been retreating in the last several decades, glacier advancing has been ubiquitously observed either on surge-type glaciers or on those where ice-rock avalanches have occurred (Paul, 2019; Schneider et al., 2011). Kääb et al. (2021a) suggested that glacier detachments could be seen as extreme endmembers of the range of surge-type and surge-like glacier instabilities, supported by the fact that some of glacier detachments exhibited surge-like advance ahead of the failures or occurred on surge-type glaciers themselves. In addition to this specific dynamic pattern, some communal geomorphic conditions have also been summarized from a compilation of 19 actual or possible glacier detachment events (Kääb et al., 2021a). The most frequent geomorphic characteristics on these detached glaciers are found to be the presence of abundant weak bedrocks/fine sediments under a glacier and gentle surface slope ranging between about 5° and 20°. As discussed in Section 5.2, both the dynamic and geomorphic patterns of the KLP-37 glacier align with these previously identified common conditions for a potential glacier detachment.

Our site-specific study of the KLP-37 glacier also adds the diversity of regional conditions for identifying avalanche-prone glaciers. First, we found that the specific shape of KLP-37 presumably plays a key role in influencing the dynamics of the glacier tongue. The narrowing of the glacier tongue results in the accumulation of large glacier masses at the glacier front, thus increasing compressive pressure. This indicates that the unique glacier geometry modulated by local topography should be considered in identifying and assessing glacier detachment hazards. Second, the KLP-37 glacier is located in a region where a few strong earthquakes have occurred in history (see Fig. 1 for the avalanches triggered by the 2001 earthquake). Given the triggering effect of large earthquakes on glacier detachment, particular attention should be paid to destabilized low-angle glaciers in active tectonic zones.

We have shown that using multi-source remote sensing images enables us to address the vertical and horizontal dynamics of a glacier, which is particularly helpful for identifying detachment-prone glaciers. In the future, monitoring techniques with short

temporal sampling rates such as ground-based SAR and optical camera-based systems should be employed to capture the transient geomorphic changes of the glacier. Particular attention should be paid to the critical signs pointing to further destabilization, such as the further acceleration of the glacier front, up-glacier growth of the fast-moving zone, additional surface crevassing, and appearance of shear margins along the edges. Our simulations of the runout extent using avalanche modeling, combined with the empirical estimates of runout distance using the angle of reach, provide a preliminary assessment of the hazard influence of a potential glacier detachment. We highlight that such assessment should be valued in the future because it is mostly the only way to give first-hand information on the possible glacier detachment influence.

## 6. Conclusions

In this study, we analyzed the multi-decadal geomorphic changes of a small low-angle valley glacier KLP-37 in the East Kunlun Mountains with multi-source remote sensing imagery, followed by a hazard assessment of the glacier. We found that the glacier tongue has undergone slow surge-like processes in the past four decades. The glacier snout had been progressively advancing during the observation period, with a total advance of about 418±24.13 m. The glacier surface exhibited continuous thinning in the source region and thickening in the tongue. Negative volume changes were found over the glacier tongue region, indicating continuous loss of the glacier mass there. We observed acceleration of the flow velocity over the glacier tongue, with the mean velocity below 4800 m more than tripled, during the period 2009–2020.

Our observations suggest that several factors control the dynamics of the KLP-37 glacier. The change of flow direction of the glacier at an elevation of about 4800 m due to the local topography, coupled with the "V" shape of the glacier tongue geometry, presumably leads to large ice mass and stress accumulating at the glacier front and plays a crucial role for the surge-like behavior. The presence of an ice-dammed glacier lake and a supraglacial pond on the glacier tongue surface implies a hydrological influence on the glacier dynamics as well. However, the mechanisms of the hydrological effects are not clear. Furthermore, the long-term climate warming and increased annual precipitation likely intensify the glacier's dynamics, as manifested by the accelerations of snout advance and surface flow during the past decades.

The runout hazard assessments from both calculations based on the angle of reach (Fahrböschung) and Voellmy-Salm modeling suggest that the avalanche of the whole KLP-37 glacier would easily reach the Qinghai-Tibet railway. However, whether the detachment of the glacier tongue would threaten the safety of the railway depends on the selection of mobility index ("Fahrböschung") or friction parameters. It is thus essential to monitor the dynamics of the KLP-37 glacier continually in the future to ensure the operation safety of the Qinghai-Tibet railway and highway downstream.

This study also demonstrates the possibility of using multiple remotely-sensed datasets to investigate the multi-decadal geomorphic changes of glaciers in mountainous regions, where direct observations are scarce. Moreover, we have presented a means of evaluating a destabilized glacier's runout hazard based on remote sensing observations. The approach presented here for the KLP-37 glacier can be easily adapted for other similar mountain glaciers in vast regions to assist in detachment hazard prevention and mitigation.

## Code availability

The open-source ASP software for generating the ASTER DEM based on the ASTER stereo images is available at https://ti.arc.nasa.gov/tech/asr/groups/intelligent-robotics/ngt/stereo/. The HEXIMAP toolbox for extracting DEM from the declassified Hexagon KH-9 satellite imagery can be downloaded at https://github.com/gmorky/heximap. The MASSFLOW software for conducting the glacier avalanche modeling is freely available at http://www.massflow-software.com/ for an educational purpose.

## Data availability

The SRTM-C and Hexagon KH-9 images are downloaded from the United States Geological Survey (USGS) EROS Archive. The SRTM-X and TanDEM-X DEMs are copyrighted and provided by the German Aerospace Center. The HMA DEMs are downloaded from NASA National Snow and Ice Data Center Distributed Active Archive Center. The ASTER stereo images are downloaded from the NASA Earthdata Search archive. The Planet images are freely downloaded from the Planet website (https://www.planet.com/markets/education-and-research/) for scientific research purposes.

## Author contribution

XW collected the satellite data, did most of the result analyses, and wrote the draft. LL and YH helped revise the manuscript. QL helped interpret the results. RZ and BZ helped pre-process some of the satellite data. TW, LZ, and GL supported the field trip to the study site in 2016. All the authors were involved in the editing of the manuscript.

## Competing interests

The authors declare that they have no conflict of interest.

## Acknowledgements

The authors would like to thank the two anonymous reviewers for their constructive comments. This study is jointly supported by the National Natural Science Foundation of China (41804009 and 42071410), the National Key Research and Development Program of China (2017YFB0502700), the Project of Application Foundation of the Sichuan (China) Science and Technology (2020YJ0322, 2020JDTD0003), and The Hong Kong Research Grants Council (CUHK14303417 and CUHK14303119). Some figures in this paper were plotted using the Generic Mapping Tools (Wessel et al., 2013).

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
