# Peer review of "Progressive advance and runout hazard assessment of a low-angle valley glacier in East Kunlun Mountains from multi-sensor satellite imagery analysis"

_Natural Hazards and Earth System Sciences, 2021_

## Referee Comment (RC2)

Progressive advance and runout hazard assessment of a low-angle valley glacier in East Kunlun Mountains from multi-sensor satellite imagery analysis

Wang et al.
* * *
**General comments**
The authors present a case-study of a low-angle valley glacier that has been advancing and is potentially destabilizing, and put this in the context of recent uptick of glacier detachment observations. The paper provides a valuable contribution to the ongoing discussion of the drivers, mechanics and possible consequences of glacier detachments. It is largely well written and easy to follow, and I believe that it should be published in NHESS after addressing the following points:

My first point of criticism is the hazard assessment, specifically the claim that the maximum runout distance of a potential glacier detachment is 1.3 km. The authors come to this conclusion using moderate friction parameters in a Voellmy-Salm model. However, a simple test using the mobility index Fahrböschung indicates that if KLP-37 were to detach in a manner similar to what has been observed at Aru, Flat Creek etc., the resulting mass flow could easily reach the railroad (Fahrböschung of 7-8°, based on my rough calculations). Kääb et al., 2020 show that Fahrböschung values for all known glacier detachments are between 5° and 10°.
    Additionally, the hazard assessment was only carried out with one ice volume estimate that the authors state is likely conservative. I believe that the hazard assessment would benefit from being conducted with a range of starting volumes and upper boundaries. This, combined with a comparison of Fahrböschung values, would, I believe, paint a more honest picture of the hazard that glaciers like KLP-37 can pose, and therefore be of higher value to the scientific and hazard management community.
    Finally, I was a bit surprised to find the entire hazard assessment in the Discussion section. I think it makes up a substantial enough part of the paper that it would deserve to be presented in the Methods and Results sections. Currently it comes as a bit of a surprise to the reader.

My second point of concern are the surface flow velocity measurements. There is a discrepancy of two orders of magnitude between the terminus advance velocity and the surface flow velocities. No physical explanation that I can think of can justify such a difference.
    I suggest that this part of the analysis be redone, taking into account the uncertainties resulting from geolocation errors in the Planet data, and presented for the glacier and its surroundings.

**Specific comments**
**Title:** I suggest saying "Progressive advance and *detachment* hazard assessment …"
**L50**: "Specifically, increasing air temperatures, coupled with…" → Unclear whether you are intending this statement in a general sense or for specific cases (which ones?). Please clarify.
**L53**: Geometry changes were also documented on Flat Creek glacier, as well as several other instances (refer to Kääb et al., 2020).

**L57**: "Recent glacier collapse events on the QTP" → Have there been more than the two at Aru? If yes, please specify, otherwise maybe just refer to the Aru cases specifically.

**L59+61**: "glacier instabilities" → what kind of instabilities are you referring to here? Just glacier detachments or also ice avalanches, surges etc.? Maybe this can be formulated to be a bit more specific.

**Figure 1**: It would be nice to show the location of the weather station in this first Figure. The caption suggest that the figure shows something about the geology, when in fact only the topography and the faults ('seismic setting'?). That said, I think a geological overview would be well placed here since the authors later make a reference to fine grained rocks, but never describe the region's geology. Such a description should be added to the Study Site section. Lastly, I am not sure the yellow dots (ice avalanches triggered by the 2001 EQ) add much information here, since they are never mentioned again.

**L104**: please clarify what you mean by "pre-collapse" event

**L106**: since you just talked about both the east and the west branches in the previous sentence, it would be helpful to specify relative to which tongues the lakes are.

**L108**: "We can observe fine debris …" → how did you observe this? High resolution images are likely not enough to determine this adequately. If you have not had a chance to investigate in the field, a reference to the geology of the area (L110 "weak bedrock") would be helpful.

**Figure 2:** In this caption, as well as somewhere in the text, please note (and possibly show) the distance between the glacier and the railroad. Also, please label what the yellow polygon indicates.

**L133ff:** In this paragraph, it is not always clear which DEMs you made and which you just acquired. Did you generate any of the radar data based DEMs yourself (seems like they are all freely available products?). A little bit of clarification would be helpful here.

**L155:** define $\partial$t

**L157:** "which is set to zero because all the optical images used are orthorectified and georeferenced products" is not a satisfying reason for setting that value to zero. In my experience, orthorectified Planet images can exhibit a lot of jitter. Since you have your cross-correlation pipeline running in MicMac, you should be able to get an estimate of this value from the off-glacier cross correlation results.

**Figure 3:** The Nuth and Kääb approach for co-registering DEMs is pretty standard – the image could easily go into the supplementary material.

**L194ff**: I'm not sure that I agree with the authors approach to the applying single offsets here. In terms of snow penetration, if I had an estimate of penetration error with elevation, I think I would chose to correct it based on a fitting a curve to the data. The authors state that they are not interested in the high reaches of the glacier, but then show such results in Fig. 6. It would be interesting to see a bit more discussion of the effects of these error sources in the discussion. Do they significantly change the amount of thinning/thickening that we can see on the different parts of the glacier?

**L215:** I don't understand what you mean by "standard deviation of the mean elevation change of its elevation band"? Aren't you just using the standard deviation of each elevation band (as indicated in equation 3?).

**L223:** Please add a reference to the least-squares fit … isotropic variogram method, or describe in more detail.

**L231:** Why only three pairs? Processing all possible pairs would make your results more robust.

**L251:** It is not clear to me what "The snowpack downstream of the crevasses collapsed" means. Was there an avalanche? Or just different snow conditions over a crevassed area? Or a change to the actual glacier ice? Please clarify this point.

**L254:** If it stayed unstable for four decades, does that mean it is actually stable? Clarify the point you are trying to make with this statement.

**Figure 4:** It is very hard to see much in this image, but there is a lot of empty space! I suggest rotating the image and making it 3x4 or 4x3 cells, so that it is easier to see something.

**L267:** Did you determine the uncertainties of the area in the same way that you did for the advance velocities, or some other method?

**Figure 5:** It would be really nice if each dot also had a horizontal bar that indicated over what time span the velocity was calculated.

**L279:** Clarify that you DON'T mean that the velocities were similar between 2009 – 2015 and 2015 – 2019, but instead were stable during those periods, but then jumped between 2015 and 2016.

**Figure 6:** Please make the colorbars a little bit larger and use the same scale for each image (in which case you can get away with just one colorbar for each line of plots). It is very hard to interpret the changes when the scales change.

**L295:** How much could this assessment be influenced by the radar penetration depths?

**L298:** over the whole east branch? Or another window? Please clarify.

**L299:** did the east branch only thin or also retreat?

**L305:** I'm not sure I agree with the term *bulging* for what is described here (and depicted in Figure 7b): Bulging implies that there was an upward motion of ice surface, but really there is just advance of ice. The DEM difference is mostly against previously un-glacierized terrain, so obviously this appears as thickening in the DEM difference, but it does not reflect thickening of ice in a place where the was ice previously. That said, I do see bulging Figure 6, but am having a hard time finding that in the cross-section.

**L306:** What makes it not a surge, but only surge-like? This could be an interesting topic to elaborate on in the discussion.

**Figure 7:** Can you indicate the years next to the different DEMs? That would help interpret the changes. Then, it would be nice if the colors were on a gradient scale along age, so that it is visually clear which way the changes progressed.

**Figure 8:** Because the x-axis runs from upglacier to downglacier in the previous plots, I think it would be beneficial to flip the x-axis in this plot (so that higher elevations are on the left and lower elevations are on the right).

**L334:** Over what area where these maximum velocities measured? Single points?

**Figure 9:** Please use colorbars that are not divergent (the blue to white to red implies that blue has an opposite signal to the red) but rather continuous, and use the same scale in each plot. Additionally, it would be very nice to see the velocities outside the glacier as well. Is there actually enough accuracy to distinguish the glacier from the surrounding landscape? Lastly, you keep referring to the 4800m elevation line, so it would be very nice to show that line in the plots.

**L349:** Your surface flow velocities and terminus advance velocity differ by two orders of magnitude. I don't see the explanations you offer as plausible, but suspect an error in the processing of the cross-correlation.

**L370:** How did you (or Wu et al?) determine the lower permafrost altitude in the field? It would also be good to take a look at the two global permafrost maps (Gruber et al., 2012, Obu et al., 2019), and reference those.

**L374:** If my memory serves me correctly, Leñas glacier was deemed temperate… but I think something similar was found at Flat Creek. Maybe double check the references?

**L379:** The statement that the ice-dammed lake exerts a hydrological influence on the glacier tongue needs to be backed up significantly. What kind of influence? This seems like it contradicts the cold-ice edge that is keeping the lake locked in. If the lake has not decreased in size, I don't see how it could influence the dynamics of the glacier in any substantial way.

**L384:** I don't understand what you mean by "deflection region"

**L386:** It seems to me that the narrowing of the topography which kind of "pinches" the glacier tongue would help stabilize it, rather than making it unstable. Please clarify how you think this factor contributes to a destabilization.

**Figure 11:** Why are you only showing summer temperatures? At what elevation are these temperatures? The current graph does not support your claim that the area is in permafrost.

**L395:** I am a bit hesitant about this interpretation of the influence of temperature and precipitation on the flow speeds. Firstly, what exactly are you referring to by "flow velocity"? The advance of the glacier tongue or the mapped velocities on all of the tongue? This needs to be clarified. For the advance of the glacier tongue, the change really just happened between 2015 and 2016, but was likely the result of accumulation that happened further upstream in the years before. Please elaborate on these points.

**Figure 12:** Why are you only considering the tongue in you hazard assessment? The crevasses seem to be much higher on the glacier, and the situation at Aru showed that that was where the detachment initiated. At the very least, I believe that it would be valuable to run the model with the two endmember volumes.

**L435:** Can you get that much advance by just internal deformation? I don't know the answer, but feeling like there might have to be some sliding, at least in the fastest parts.

**L458:** I don't see how the current glacier flow velocities / basal friction parameters influence how far the mass flow resulting from a glacier detachment can travel. If the entire thing detaches, basal friction decreases to essentially nothing, and that is what determines the runout distance.

**L461:** I am not convinced by your justification for selecting the moderate friction parameters. I am not sure exactly what findings you are referring to with reference to the Allen 2009 paper, but until very recently, we had hardly heard about glacier detachments, and it is unlikely that a comprehensive assessment of basal friction values was published over 10 years ago. Furthermore, I am not aware of any findings that link the basal shear stress during regular glacier flow to the runout distances of glacier detachments, most certainly not for making a difference between surging and non-surging glaciers. Please clarify substantially.

**L474:** I don't understand what you mean by "experiencing ice-rock collapses"?

**L475:** I don't follow how Kääb 2020 suggest that any glacier advance is an indicator for a glacier instability (though I suppose this depends on how you define glacier instability – not all are hazardous).

**L481:** Please back up the statement of "the ice-dammed lake influencing the dynamics of the glacier tongue of the west branch". This is currently not supported by any of the data, nor has any relationship between the two been mentioned prior to this.

**L487:** Please back up the statement "was transported from the upper region by a historical seismic event". How did you determine this to be a logical possibility? Is there evidence of something like this elsewhere in the region? So far, we have not seen earthquakes causing detachments of low-angle valley glaciers… (not saying it's not possible though…).

**L492:** In addition to ground based InSAR, optical camera based systems are probably cheaper and similarly effective, if it's just for monitoring.

**L494:** I don't understand what you mean by saying "our simulations provide an alternative solution for assessing the hazard of an impending glacier collapse". Alternative to what? Is the glacier detachment impending, or just a possibility (I would argue that it is probably more likely that nothing will happen).

**Technical corrections**

Terminology: Over recent months, the term *glacier detachment* seems to have become the term of choice for describing the catastrophic detachment of low-angle valley glaciers. Rather than using the somewhat fuzzy term "collapse", which is frequently used for rock avalanches, slope failures etc., I recommend changing the terminology to glacier detachment throughout the manuscript. Example: Due to hazardous threats of glacier  detachments to … (Line 37).

Use of tenses: There are quite a few improper uses of past tenses. I have noted these below where I caught them, but am likely to have missed a few. In general, I suggest to put anything that the authors have done in the past tense. Example: We chose a set of parameters, we analyzed these images etc.

If your line numbers have changed, I also have these edits in a document (hand written). Please let me know via the editorial office if having this document would be helpful.

L26: *tripled* not trebled
L39: *glacier* not glacial
L40: *was* not have
L41: *was* not is
L46: *two* not twice
L46: remove *the lower parts* → the entire glacier was implicated in 2015
L53: *on* or *from* not in
L59: *mean global rate*, not global mean rate
L67: …identified the glacier, *which is* close to …
L68: intense *crevassing* on the glacier surface *raised* the question *whether* a hazardous ice avalanche *might be* imminent.
L71: *past* not recent
L75: *discuss* not discussed
L76: *estimate* not estimated
L77: *occurred* not occurs; *discuss* not discussed
L104: the west branch's *terminus lies about 220m lower* (Fig. 2a); remove "Particularly"
L112: *overlayed* not overlapped; *The* ice-dammed lake and supraglacial pond are also *shown*.
L118: Why are Table 1 and Table 2 shown at the end of the document? It would be much nicer to have them where they are referenced.
L121**:** I would say that glacier changes are *often* mapped on Landsat etc. images (partly bc for decades that is the best we had) and that they have a *resolution of 10-30m*.
L123: *used* (if you got it free) or *acquired* (if you bought the images) not "collected" (collected is used primarily for data collection that you do in the field). This comes up several times in the manuscript.
L127: Just say …*spy satellite to reconstruct the topography of KLP-37 in 1975*.
L161: *generate* DEM*s from* KH-9 stereo images
L177: *to* a spatial *posting* of; The DEM pairs *need* to be …
L185: *posting* or *resolution* not post
L196: odd use of while
L200: similar to  previous
L248: insert space after KH-9; *past* instead of recent
L249: *A time-lapse* of optical images …
L251: Remove "KH-9 image" since the crevassing didn't happen in the image, but rather in the world. The images just show it.
L262: *is* instead of were

L277: *the past* instead of recent

L278: *and* instead of while; *rose to more* instead of had been higher

L341: *tripled* not trebled

L350: remove one *not*

L360: *advanced continuously* instead of "had been progressively advancing"

L361: *accelerated* instead of was accelerating

L362: remove *temporal*

L363: …conditions of *the glacier,* topography …

L371: *terminus* instead of termini

L377: … crevasses *to reach the* sliding surface …

L403: *dense* is an odd choice of word here – what do you mean?

L403: *was* instead of "has been"; I don't understand what you mean by "within 40 days before"? 40 days prior? Or during the 40 days prior? Or up to 40 days prior?

L501: with *a total* advance instead of "with an accumulative distance"

---

## Author Comment (AC1)

**Reply to the Comments of Referee #1**

We wish to thank referee #1 for the helpful and constructive comments on our manuscript (Manuscript No.: nhess-2021-57). We have revised the manuscript carefully to address the issues. Our item-by-item responses to the comments are provided below.

This is over large parts a sound and thorough study about the interesting unusual slow surge-type motion of a small glacier on the Tibet Plateau. The manuscript is well written. I recommend publication in NHESS after consideration of the following remarks, some of them substantial.

Reply: Thank you very much for you positive comments.

SUBSTANTIAL COMMENTS

A. The glacier velocities need further work and explanation:

i. I recommend very much to extend the velocity measurements to 2021. There seems to be much continued activity during this period (see also below).

Reply: Good suggestion! We have added two cloud-free Planet images acquired on 2020/08/29 and 2021/03/23 to extend the velocity measurements. The figures showing the glacier snout advance velocity and tongue flow velocity have also been updated. Note that the image 2021/03/23 was not used for flow velocity measuring because the glacier tongue was partly covered by snow at the time.

ii. The fact that advance velocities seem larger than surface velocities is puzzling. Your measurement grid is so dense that I don't believe your explanation 2 (L 350 and following) can be right (local underestimation of displacement). I do not understand fully your explanation 1. It could be that the very terminus lowers and slides out. You would have to assess closer if this is able to explain the magnitude of advance observed.

Reply: Sorry for the puzzling explanations on the flow velocities. We have checked the image correlation processing and found that we initially gave wrong flow velocity estimates. We have redone the cross-correlation using MicMac with multiple parameter combinations. We found that we can obtain reasonable flow velocity measurements by specifying a privileged direction for regularization (i.e., the main flow direction) when doing the correlation. We have also verified the velocity estimates by implementing the image cross-correlation using the ImGRAFT software (Messerli & Grinsted, 2015). The renewed correlation results show that the glacier tongue moves at a maximum velocity of about 30 m·a$^{-1}$ in recent years, comparable to the snout advance velocity inferred from visual interpretation. We have updated the results and descriptions for the flow velocity field of KLP-37 in the revised paper.

iii. I checked the Planet images you used for the velocities, and I am little convinced by your results. Over 2009-2012, there are large distortions between the images. How did you correct for them? For the other periods it is very difficult to follow features over 2-4 years as they are changing too much. Visually, I manage to follow features in a good way only over 1-year periods. If I compare (visually) 1-year periods, I can well see that only the lowermost part of the tongue well below the lake is accelerating. You will have to thoroughly redo the velocity measurements, I guess.

Reply: Thanks a lot for the insightful comments and good suggestion. As stated above, we have redone the image correlation processing with Planet images. The cross-correlations have been conducted on image pairs with one-year separation in the revised paper. We have also added the velocity estimate between 2019/08/06-2020/08/29.

B. The authors shortly touch upon permafrost, but not on creeping permafrost landforms. Obu et al. (2019) model mean annual ground temperatures for the glacier tongue area of -4 - -6 deg

Reply: Yes, we inferred that the glacier tongue is located in a permafrost environment. We have checked the permafrost map based on TTOP modeling by Obu et al. (2019), and have found that the permafrost probability value over the KLP-37 glacier region is 1. We have clarified this in the revised manuscript. We indeed did not touch more about creeping permafrost landform in the initial version of the paper. We have added some discussions on whether the landform should be interpreted as an active rock glacier or ice-code moraine, according to your suggestion below (comments #C)

C. The glacier front has a sharp steep front, reminding of rock glaciers. So, the acceleration of the tongue should also be seen in terms of the dynamics ice-cored moraines and rock glaciers. Both are known to be able to show collapse-like behavior. In particular rock glaciers have recently been shown to accelerate and collapse. This aspect needs to be discussed in the paper.

Reply: Thanks for the insightful comments. In the discussion part (Section 5.1), we have explained why the landform in the front of KLP-37 is part of the glacier itself and not a rock glacier or ice-cored moraine.

The KLP-37 glacier tongue differs from a rock glacier both in morphology and kinematic pattern. (1) Rock glaciers are landforms consisting of mixtures of unconsolidated rock debris and ice in an alpine environment. The debris cover on a rock glacier is usually thick (>3 m), and the ice content is lower than 45% (Janke et al., 2015). Compared to the debris-covered glacier, rock glaciers' debris cover is less homogenous and coarser. The field photo in 2016 (Fig. 2c) shown that the debris cover on the KLP-37 glacier tongue was thin and uniform. The exposed clean ice in the photo also indicated rich ice content of the landform. (2) On the surface of the KLP-37 glacier front, we did not observe distinctive ridges and furrows, which are distinctive characteristics of rock glaciers. (3) Rock glaciers

typically move downslope at velocities smaller than 10 m·a⁻¹ (Delaloye et al., 2010; Kääb et al., 2021), while the flow velocity from our cross-correlation measurement reaches a maximum velocity of ~30 m·a⁻¹.

Ice-cored moraines are ice-marginal landforms that comprise a discrete body of glacier ice buried underneath sediment (Singh et al., 2011). Ice-cored moraines are generally formed by the isolation of a body of glacier ice through the establishment of a sediment/debris cover near the margin, which, if sufficiently thick, shields the ice from melting. The different melting between the protected sediment-covered ice and clean ice up glacier then often results in the sediment-covered ice body cut off from the supply of active pure ice. Therefore, a typical ice-cored moraine should be disconnected from the active glacier ice margin. However, some authors prefer to use the term much more loosely to include moraine-like ridges in supraglacial sediment underlain by ice, irrespective of the fact that the ice s still flowing (active) and continuous underneath the sediment cover (i.e., Lønne & Lyså , 2005; Evans, 2009).

Here we follow the strict definition that ice-cored moraine is a landform that (1) is disconnected from the active glacier ice margin and (2) contains a discrete body of ice that is surrounded by sediment (Singh et al., 2011). Regarding the front of the KLP-37, the long-term advance of the glacier front without cut-off indicates the inner body is connected with the glacier margin. Therefore, we suggest that the landform we investigated is part of the KLP-37 glacier tongue covered by a thin debris layer.

References

Evans, D.J.A.: Controlled moraines: origins, characteristics and palaeoglaciological implications, Quaternary Science Reviews, 28, 183–208, 2009.

Janke, J.R., Bellisario, A.C., and Ferrando, F. A.: Classification of debris-covered glaciers and rock glaciers in the Andes of central Chile, Geomorphology, 241, 98–121, 2015.

Lønne, I., and Lyså, A.: Deglaciation dynamics following the Little Ice Age on Svalbard: implications for shaping of landscapes at high latitudes, Geomorphology, 72, 300–319, 2005.

Singh, V. P., Singh, P., and Haritashya, U. K.: Encyclopedia of snow, ice and glaciers, 2011.

Obu, J., Westermann, S., Bartsch, A., Berdnikov, N., Christiansen, H. H., Dashtseren, A., Delaloye, R., Elberling, B., Etzelmuller, B., Kholodov, A., Khomutov, A., Kaab, A., Leibman, M. O., Lewkowicz, A. G., Panda, S. K., Romanovsky, V., Way, R. G., Westergaard-Nielsen, A., Wu, T. H., Yamkhin, J., and Zou, D. F.: Northern Hemisphere permafrost map based on TTOP modelling for 2000-2016 at 1 km² scale, Earth-Sci Rev, 193, 299-316, https://doi.org/10.1016/j.earscirev.2019.04.023, 2019.

Some random relevant papers on rock glacier acceleration and collapse:

Delaloye, R., Lambiel, C., and Gärtner-Roer, I.: Overview of rock glacier kinematics research in the Swiss Alps: Seasonal rhythm, interannual variations and trends over several decades, Geographica Helvetica, 65, 135-145, https://doi.org/10.5194/gh-65-135-2010, 2010.

Kääb, A., Strozzi, T., Bolch, T., Caduff, R., Trefall, H., Stoffel, M., and Kokarev, A.: Inventory and changes of rock glacier creep speeds in Ile Alatau and Kungöy Ala-Too, northern Tien Shan, since the 1950s, Cryosphere, 15, 927-949, https://doi.org/10.5194/tc-15-927-2021, 2021.

Bodin, X., Krysiecki, J.-M., and Iribarren-Anacona, P.: Recent collapse of rock glaciers: two study cases in the Alps and in the Andes, 12th INTERPRAEVENT, Grenoble, 2012,

Kofler, C, Mair, V, Gruber, S, et al. When do rock glacier fronts fail? Insights from two case studies in South Tyrol (Italian Alps). Earth Surf. Process. Landforms. 2021; 1– 17. https://doi.org/10.1002/esp.5099

Kääb, A., Frauenfelder, R., and Roer, I.: On the response of rock glacier creep to surface temperature increase, Global Planet Change, 56, 172-187, https://doi.org/10.1016/j.gloplacha.2006.07.005, 2007.

… and many others that are cited in the above.

**SPECIFIC COMMENTS**

L 105: To suggest that the lower terminus is a sign of an earlier collapse is quite speculative and should not appear in the study site description. In addition, I doubt you can draw this conclusion. The debris-covered lower part of the glacier might be an ice-cored moraine as there are many found in the region. I cannot see how this feature is so different from the other ice-cored moraines in the region that you can suggest it might be from a collapse.

Reply: We have removed the statement in the study site description. We have added some discussions to explain why we interpreted the KLP-37 front as part of the glacier tongue.

L 129: scanned at 7 mm is certainly wrong. 7 micrometers? Please check the USGS pages

Reply: Yes, the KH-9 images were scanned at a resolution of 7 micrometers. We have corrected it in the paper.

Fig. 3 could perhaps go into the Supplement. NHESS readers are less interested in such technical details as they may distract from the hazards aspects. If you agree, move lines 186-193 to the Suppl.

Reply: Good suggestion. We have moved Fig. 3 and the relevant descriptions (i.e., Lines 186-193) to the Supplementary file.

L 196: something wrong with the date 2104/10/18. 2014?

Reply: Thanks for pointing out this type error, and we have corrected it to 2010/12/05.

L 202: could be interesting to add the C-band – X- band differences in the Supplement. Did you apply the 2.82 m correction also on the debris-covered lowest part of the glacier? I doubt the penetration will be 2.8 m through a debris layer.

Reply: We have added a figure showing the elevation difference between the C- and X-band DEMs over the KLP-37 glacier in the Supplementary file. The penetration depth indeed varies with the changes of elevation (see Fig. R1 below). Note the DEM difference values >±10 m were defined as outliers and were removed. Statistics show that the mean elevation differences were about 0.43 m and 2.36 m for regions below and above 5000 m, respectively. We have followed the suggestion of reviewer #2 by choosing a curve fitting to the elevation differences and then applied the correction with the fitted function.

[Figure]

Fig. R1. Plot of surface elevation difference between SRTM C-band and X-band DEMs against elevation. The red squares indicate mean values of elevation differences for elevation bins with a 100 m separation.

L233: Why do you stop matching surface displacement in 2019. From a quick check of the Planet archive I see that displacements 2019-2020 and 2020-2021 would be very interesting to describe the instability. I strongly recommend to update the velocities after 2019. I almost consider that mandatory for the purpose of your work.

Reply: We have added two cloud-free Planet images acquired on 2020/08/29 and 2021/03/23 to extend the velocity measurements. The figures showing the glacier snout advance velocity and tongue flow velocity have also been updated. Note that the image 2021/03/23 was not used for flow velocity measuring because the glacier tongue was partly covered by snow at the time.

L 254: the question is if a 40-year continuous development should be called "unstable".

Reply: We have modified the sentence to "The highly developed crevasses in the glacier accumulation region since the 1970s and the widening of the crevasses in recent years indicate that the glacier may develop toward destabilization."

L 261: the lake *was* not visible …

Reply: We have replaced the word "were" to "was".

L 306: Calling a 40-year mass displacement "surge-like" might be open to discussion. How about calling it "… and imply a slow surge-like mass transfer process in the tongue area …"? See above, where I suggest though from checking repeat Planet images that only the lower most part of the tongue is actually accelerating. I would not call that surge-like at all. It seems more a slow landslide of the frontal ice-cored moraine, or similar.

Reply: Good suggestion. We have modified the sentence to "Elevation changes confirm the stepwise

glacial termini advance (see Section 4.1) and imply a slow surge-like mass transfer process in the tongue area". Here we use the word "surge-like" according to the flow velocity patterns of KLP-37. The advance velocity of the glacier snout shown an abrupt increase during 2015 and 2016, while the rates before 2015 and after 2016 were relatively stable. This kind of flow behavior resembles glacier surging (Kääb et al., 2018) and has also been reported on several detached glaciers preceding ice avalanches, such as the Aru and Amney Machen glaciers in Tibetan Plateau (Paul, 2019; Kääb et al., 2021).

L 341: trebled –> tripled ?

Reply: Corrected.

L 342: "unstable"? see above, not sure this is destabilization?

Reply: The sentence has been modified to "The almost tripled mean velocity below 4800 m in the recent decade suggests that the glacier tongue has been getting more active towards destabilization."

L 350 and following: the two reasons why the advance should be faster than the surface velocities are not convincing to me. This effect is quite unusual and seemingly violates physical laws of flow/creep.

Reply: We have re-done the image correlation processing and updated the velocity field from 2009 to 2020.

L 411. Section 5.2 is to a large extent methods and results. I recommend to describe the model in the method section, and the model outcome in the results. Only the discussion of the model results (different parameter settings; not whole glacier collapse modelled, etc.) would then come in the discussions.

Reply: Good suggestion. We have split Section 5.2 following the suggestion. We have moved the modeling descriptions into the Method part and modeling results into the Results section. In Section 5.2, we discussed possible factors that may influence the modeling results.

L 413: changing ice flow direction: do you mean a change over time (I don't find that in the results) or a change in direction along the glacier (spatial change)?

Reply: Here we mean the change of flow direction in the spatial domain. We have clarified that: "the ice flow directions along the two glaciers have both changed due to the local topography."

L 463: … an avalanche from …

Reply: Corrected.

END OF REVIEW

---

## Author Response (AR1)

**Reply to the Comments of Referee #1**

We wish to thank referee #1 for the helpful and constructive comments on our manuscript (Manuscript No.: nhess-2021-57). We have revised the manuscript carefully to address the issues. The changes in the manuscript are highlighted with blue color. Our item-by-item responses to the comments are provided below.

This is over large parts a sound and thorough study about the interesting unusual slow surge-type motion of a small glacier on the Tibet Plateau. The manuscript is well written. I recommend publication in NHESS after consideration of the following remarks, some of them substantial.

Reply: Thank you very much for you positive comments.

SUBSTANTIAL COMMENTS

A. The glacier velocities need further work and explanation:

i. I recommend very much to extend the velocity measurements to 2021. There seems to be much continued activity during this period (see also below).

Reply: Good suggestion! We have added two cloud-free Planet images acquired on 2020/08/29 and 2021/03/23 to extend the velocity measurements. The figures showing the glacier snout advance velocity and tongue flow velocity have also been updated. Note that the image 2021/03/23 was not used for flow velocity measuring because the glacier tongue was partly covered by snow at the time.

ii. The fact that advance velocities seem larger than surface velocities is puzzling. Your measurement grid is so dense that I don't believe your explanation 2 (L350 and following) can be right (local underestimation of displacement). I do not understand fully your explanation 1. It could be that the very terminus lowers and slides out. You would have to assess closer if this is able to explain the magnitude of advance observed.

Reply: Sorry for the puzzling explanations on the flow velocities. We have redone the cross-correlation using MicMac. We found that we can obtain reasonable flow velocity measurements by specifying a privileged direction for regularization (i.e., the main flow direction of the glacier) when doing the correlation. The renewed correlation results show that the glacier tongue moves at a maximum velocity comparable to the snout advance velocity inferred from visual interpretation. We have updated the results and descriptions for the flow velocity field of KLP-37 in the revised paper.

iii. I checked the Planet images you used for the velocities, and I am little convinced by your results. Over 2009-2012, there are large distortions between the images. How did you correct for them? For

the other periods it is very difficult to follow features over 2-4 years as they are changing too much. Visually, I manage to follow features in a good way only over 1-year periods. If I compare (visually) 1-year periods, I can well see that only the lowermost part of the tongue well below the lake is accelerating. You will have to thoroughly redo the velocity measurements, I guess.

Reply: Thanks a lot for the insightful comments and good suggestion. As stated above, we have redone the image correlation processing with Planet images. The cross-correlations have been conducted on image pairs with one-year separation in the revised paper. We have also added the velocity estimate between 2019/08/06–2020/08/29.

B. The authors shortly touch upon permafrost, but not on creeping permafrost landforms. Obu et al. (2019) model mean annual ground temperatures for the glacier tongue area of -4 - -6 deg

Reply: Yes, we inferred that the glacier tongue is located in a permafrost environment. We have checked the permafrost map based on TTOP modeling by Obu et al. (2019), and have found that the permafrost probability value over the KLP-37 glacier region is 1. We have clarified this in the revised manuscript. We indeed did not touch more about creeping permafrost landform in the initial version of the paper. We have added some discussions on whether the landform of concern should be interpreted as an active rock glacier or ice-cored moraine, according to your suggestion below (comments #C)

C. The glacier front has a sharp steep front, reminding of rock glaciers. So, the acceleration of the tongue should also be seen in terms of the dynamics ice-cored moraines and rock glaciers. Both are known to be able to show collapse-like behavior. In particular rock glaciers have recently been shown to accelerate and collapse. This aspect needs to be discussed in the paper.

Reply: Thanks for the insightful comments. In the discussion part (Section 5.1), we have explained why the landform in the front of KLP-37 is part of the glacier itself and not an ice-cored moraine or rock glacier (Section 5.1).

Ice-cored moraines are generally formed by the isolation of a body of glacier ice through the establishment of a sediment/debris cover near the margin, which, if sufficiently thick, shields the ice from melting (Lukas, 2011). The different melting between the protected sediment-covered ice and clean ice up glacier then often results in the sediment-covered ice body cutting off from the supply of active pure ice. Therefore, a typical ice-cored moraine should be disconnected from the active glacier ice margin. However, some authors prefer to use the term much more loosely to include moraine-like ridges in supraglacial sediment underlain by ice, irrespective of the fact that the ice s still flowing (active) and continuous underneath the sediment cover (i.e., Lønne & Lyså , 2005; Evans, 2009).

Here we follow the strict definition that ice-cored moraine is a landform that (1) is disconnected from the active glacier ice margin and (2) contains a discrete body of ice that is surrounded by sediment

(Lukas, 2011). Regarding the front of the KLP-37, the long-term advance of the glacier front without cutoff indicates the inner body is connected with the glacier margin. We thus suggest that the landform we investigated is not an ice-cored moraine.

The KLP-37 glacier tongue also differs from a rock glacier both in morphologic and kinematic patterns. (1) Rock glaciers are landforms consisting of mixtures of unconsolidated rock debris and ice in an alpine environment. The debris cover on a rock glacier is usually thick (>3 m), and the ice content is lower than 45% (Janke et al., 2015). Compared to the debris-covered glacier, rock glaciers' debris cover is less homogenous and coarser. The field photo in 2016 (Fig. 2c) shown that the debris cover on the KLP-37 glacier tongue was thin and uniform. The exposed clean ice in the photo also indicated the rich ice content of the landform. (2) On the surface of the KLP-37 glacier front, we did not observe distinctive ridges and furrows, which are distinctive characteristics of rock glaciers. (3) Rock glaciers typically move downslope at velocities smaller than 10 m·a$^{-1}$ (Delaloye et al., 2010; Wang et al., 2017; Kääb et al., 2021b), while the flow velocity from our cross-correlation measurement reaches a maximum velocity of ~30 m·a$^{-1}$. We thus conclude that the landform of concern is not a rock glacier.

References

Evans, D.J.A.: Controlled moraines: origins, characteristics and palaeoglaciological implications, Quaternary Science Reviews, 28, 183–208, 2009.

Janke, J.R., Bellisario, A.C., and Ferrando, F. A.: Classification of debris-covered glaciers and rock glaciers in the Andes of central Chile, Geomorphology, 241, 98–121, 2015.

Lønne, I., and Lyså, A.: Deglaciation dynamics following the Little Ice Age on Svalbard: implications for shaping of landscapes at high latitudes, Geomorphology, 72, 300–319, 2005.

Lukas, S.: Ice-cored moraines, In: Singh, V., Singh, P., and Haritashya, U.K. (Eds.), Encyclopedia of Snow, Ice and Glaciers. Springer, Heidelberg, 616-619, 2011.

Wang, X., Liu, L., Zhao, L., Wu, T., Li, Z., and Liu, G.: Mapping and inventorying active rock glaciers in the northern Tien Shan of China using satellite SAR interferometry, The Cryosphere, 11, 997–1014, doi:10.5194/tc-11-997-2017, 2017.

Obu, J., Westermann, S., Bartsch, A., Berdnikov, N., Christiansen, H. H., Dashtseren, A., Delaloye, R., Elberling, B., Etzelmuller, B., Kholodov, A., Khomutov, A., Kaab, A., Leibman, M. O., Lewkowicz, A. G., Panda, S. K., Romanovsky, V., Way, R. G., Westergaard-Nielsen, A., Wu, T. H., Yamkhin, J., and Zou, D. F.: Northern Hemisphere permafrost map based on TTOP modelling for 2000-2016 at 1 km$^2$ scale, Earth-Sci Rev, 193, 299-316, https://doi.org/10.1016/j.earscirev.2019.04.023, 2019.

Some random relevant papers on rock glacier acceleration and collapse:

Delaloye, R., Lambiel, C., and Gärtner-Roer, I.: Overview of rock glacier kinematics research in the Swiss Alps: Seasonal rhythm, interannual variations and trends over several decades, Geographica Helvetica, 65, 135-145, https://doi.org/10.5194/gh-65-135-2010, 2010.

Kääb, A., Strozzi, T., Bolch, T., Caduff, R., Trefall, H., Stoffel, M., and Kokarev, A.: Inventory and changes of rock glacier creep speeds in Ile Alatau and Kungöy Ala-Too, northern Tien Shan, since the 1950s, Cryosphere, 15, 927-949, https://doi.org/10.5194/tc-15-927-2021, 2021.

Bodin, X., Krysiecki, J.-M., and Iribarren-Anacona, P.: Recent collapse of rock glaciers: two study cases in the Alps and in the Andes, 12th INTERPRAEVENT, Grenoble, 2012,

Kofler, C, Mair, V, Gruber, S, et al. When do rock glacier fronts fail? Insights from two case studies in South Tyrol (Italian Alps). Earth Surf. Process. Landforms. 2021; 1– 17. https://doi.org/10.1002/esp.5099

Kääb, A., Frauenfelder, R., and Roer, I.: On the response of rock glacier creep to surface temperature increase, Global Planet Change, 56, 172-187, https://doi.org/10.1016/j.gloplacha.2006.07.005, 2007.

… and many others that are cited in the above.

**SPECIFIC COMMENTS**

L 105: To suggest that the lower terminus is a sign of an earlier collapse is quite speculative and should not appear in the study site description. In addition, I doubt you can draw this conclusion. The debris-covered lower part of the glacier might be an ice-cored moraine as there are many found in the region. I cannot see how this feature is so different from the other ice-cored moraines in the region that you can suggest it might be from a collapse.

Reply: We have removed the statement in the study site description. We have added some discussions to explain why we interpreted the KLP-37 front as part of the glacier tongue in Section 5.1.

L 129: scanned at 7 mm is certainly wrong. 7 micrometers? Please check the USGS pages

Reply: Yes, the KH-9 images were scanned at a resolution of 7 micrometers. We have corrected it in the paper.

Fig. 3 could perhaps go into the Supplement. NHESS readers are less interested in such technical details as they may distract from the hazards aspects. If you agree, move lines 186-193 to the Suppl.

Reply: Good suggestion. We have moved Fig. 3 and the relevant descriptions (i.e., Lines 186-193) to the Supplementary file.

L 196: something wrong with the date 2104/10/18. 2014?

Reply: Thanks for pointing out this type error, and we have corrected it to 2010/12/05.

L 202: could be interesting to add the C-band – X- band differences in the Supplement. Did you apply the 2.82 m correction also on the debris-covered lowest part of the glacier? I doubt the penetration will be 2.8 m through a debris layer.

Reply: We have added a figure showing the elevation difference between the C- and X-band DEMs over the KLP-37 glacier in the Supplementary file. The penetration depth indeed varies with the changes of elevation (see Fig. R1 below). Note the DEM difference values >±10 m were defined as outliers and were removed. Statistics showed that the mean elevation differences were about 0.43 m and 2.36 m for regions below and above 5000 m, respectively. We have followed the suggestion of reviewer #2 by choosing a curve fitting to the elevation differences and then applied the correction with the fitted model.

[Figure]

Fig. R1. Plot of surface elevation difference between SRTM C-band and X-band DEMs against elevation. The red squares indicate mean values of elevation differences for elevation bins with a 100 m separation.

L233: Why do you stop matching surface displacement in 2019. From a quick check of the Planet archive I see that displacements 2019-2020 and 2020-2021 would be very interesting to describe the instability. I strongly recommend to update the velocities after 2019. I almost consider that mandatory for the purpose of your work.

Reply: We have added two cloud-free Planet images acquired on 2020/08/29 and 2021/03/23 to extend the velocity measurements. The figures showing the glacier snout advance velocity and tongue flow velocity have also been updated. Note that the image 2021/03/23 was not used for flow velocity measuring because the glacier tongue was partly covered by snow at the time.

L 254: the question is if a 40-year continuous development should be called "unstable".

Reply: We have modified the sentence to "The highly developed crevasses in the glacier accumulation region since the 1970s and the widening of the crevasses in recent years indicate that the glacier may develop towards destabilization."

L 261: the lake *was* not visible …

Reply: We have replaced the word "were" to "was".

L 306: Calling a 40-year mass displacement "surge-like" might be open to discussion. How about calling it "… and imply a slow surge-like mass transfer process in the tongue area …"? See above, where I suggest though from checking repeat Planet images that only the lower most part of the tongue is actually accelerating. I would not call that surge-like at all. It seems more a slow landslide of the frontal ice-cored moraine, or similar.

Reply: Good suggestion. We have modified the sentence to "Elevation changes confirm the stepwise

glacial terminus advance (see Section 4.1) and imply a slow surge-like mass transfer process in the tongue area". Here we use the word "surge-like" because the flow velocity of KLP-37 shown stepwise increase between 1975 and 2016. This kind of flow behavior resembles the surging movement of surge-type glaciers, which has also been reported on the detached Aru and Amney Machen glaciers on the Tibet Plateau (Paul, 2019; Kääb et al., 2021a). Additionally, ice flow accelerations were also observed on the upper part of the glacier tongue from the updated cross-correlation measurements. We have made some clarifications in Section 4.2 in the revised paper.

L 341: trebled –> tripled ?

Reply: Corrected.

L 342: "unstable"? see above, not sure this is destabilization?

Reply: The sentence has been modified to "The more than tripled mean velocity below 4800 m in the recent decade suggests that the glacier tongue has been getting more active towards destabilization."

L 350 and following: the two reasons why the advance should be faster than the surface velocities are not convincing to me. This effect is quite unusual and seemingly violates physical laws of flow/creep.

Reply: We have re-done the image correlation processing and updated the velocity field from 2009 to 2020. The renewed correlation results show that the glacier tongue moves at a maximum velocity comparable to the snout advance velocity inferred from visual interpretation.

L 411. Section 5.2 is to a large extent methods and results. I recommend to describe the model in the method section, and the model outcome in the results. Only the discussion of the model results (different parameter settings; not whole glacier collapse modelled, etc.) would then come in the discussions.

Reply: Good suggestion. We have moved the modeling descriptions into the Method part and modeling results into the Results section. In Section 5.2, we discussed the limitations of the VS modeling.

L 413: changing ice flow direction: do you mean a change over time (I don't find that in the results) or a change in direction along the glacier (spatial change)?

Reply: Here, we mean the change of flow direction in the spatial domain. We have clarified that: "the ice flow directions along the two glaciers have both changed due to the local topography."

L 463: … an avalanche from …

Reply: We have rewritten the relevant sentences.

END OF REVIEW

**Reply to the Comments of Referee #2**

We want to convey many thanks to the anonymous referee #2 for the thorough and constructive comments on our manuscript (Manuscript No.: nhess-2021-57). We have revised the paper carefully following the comments. The changes in the manuscript are highlighted with blue color. Our item-by-item responses are provided below.

**General comments**

The authors present a case-study of a low-angle valley glacier that has been advancing and is potentially destabilizing, and put this in the context of recent uptick of glacier detachment observations. The paper provides a valuable contribution to the ongoing discussion of the drivers, mechanics and possible consequences of glacier detachments. It is largely well written and easy to follow, and I believe that it should be published in NHESS after addressing the following points:

Reply: Thank you very much for your positive comments on our paper.

My first point of criticism is the hazard assessment, specifically the claim that the maximum runout distance of a potential glacier detachment is 1.3 km. The authors come to this conclusion using moderate friction parameters in a Voellmy-Salm model. However, a simple test using the mobility index Fahrböschung indicates that if KLP-37 were to detach in a manner similar to what has been observed at Aru, Flat Creek etc., the resulting mass flow could easily reach the railroad (Fahrböschung of 7-8°, based on my rough calculations). Kääb et al., 2020 show that Fahrböschung values for all known glacier detachments are between 5° and 10°.

Additionally, the hazard assessment was only carried out with one ice volume estimate that the authors state is likely conservative. I believe that the hazard assessment would benefit from being conducted with a range of starting volumes and upper boundaries. This, combined with a comparison of Fahrböschung values, would, I believe, paint a more honest picture of the hazard that glaciers like KLP-37 can pose, and therefore be of higher value to the scientific and hazard management community.

Finally, I was a bit surprised to find the entire hazard assessment in the Discussion section. I think it makes up a substantial enough part of the paper that it would deserve to be presented in the Methods and Results sections. Currently it comes as a bit of a surprise to the reader.

Reply: Thanks for the insightful comments and good suggestions.

We may indeed obtain a biased hazard assessment of possible avalanche by only using the Voellmy-Salm model with empirical parameters. Calculating the maximum runout distance using the mobility index Fahrböschung (i.e., angle of reach) is a good suggestion to provide an alternative hazard assessment. In the revised paper, we have provided runout estimates from both avalanche modeling and the mobility index Fahrböschung angle. Given the upper and lower bounds of Fahrböschung angle (5°~10°) for low-angle glacier detachments suggested by Kääb et al. (2021a), we have roughly estimated the runout distance of possible avalanches from KLP-37.

We have also adopted the suggestion of conducting runout assessments by considering two endmember avalanche scenarios with different source volumes. The first scenario assumes that the detachment starts from the transverse crevasses in the glacier accumulation region, whereas the

second scenario assumes an avalanche of the glacier tongue. The avalanche volumes for the two scenarios were estimated to be about $27.06 \times 10^6$ and $6.63 \times 10^6$ m$^3$, respectively. Given moderate friction parameters of the Voellmy-Salm model, we have found that the avalanche material of scenario #1 would reach the Qinghai-Tibet railway, while scenario #2 would not. The Fahrböschung values for the two avalanche scenarios were calculated to be both about 9°, laying in the range of 5°~10°.

Following the suggestions on the paper structure, we have moved the method descriptions for hazard assessment into Section 3. The avalanching modeling results have been moved to Section 4.

My second point of concern are the surface flow velocity measurements. There is a discrepancy of two orders of magnitude between the terminus advance velocity and the surface flow velocities. No physical explanation that I can think of can justify such a difference. I suggest that this part of the analysis be redone, taking into account the uncertainties resulting from geolocation errors in the Planet data, and presented for the glacier and its surroundings.

Reply: Great suggestion. We have redone the cross-correlation processing using MicMac. We found that we can obtain a reasonable flow velocity estimate by specifying a privileged direction for regularization (i.e., the main flow direction of the glacier) when doing the correlation. In the revised manuscript, we have updated the descriptions for the flow velocity of KLP-37. The velocity estimates for the surroundings of KLP-37 are also presented.

**Specific comments**

**Title:** I suggest saying "Progressive advance and *detachment* hazard assessment ..."

Reply: Good suggestion! We have modified the title to "Multi-decadal geomorphic changes of a low-angle valley glacier in East Kunlun Mountains: remote sensing observations and detachment hazard assessment" in the revised version.

**L50**: "Specifically, increasing air temperatures, coupled with...". Unclear whether you are intending this statement in a general sense or for specific cases (which ones?). Please clarify.

Reply: We have removed this unclear statement in the revised version.

**L53**: Geometry changes were also documented on Flat Creek glacier, as well as several other instances (refer to Kääb et al., 2020).

Reply: We have rewritten the sentence as follows "The detachments of several glaciers such as the Aru and Flat Creek glaciers were also preceded by geometric changes in the form of surge-like behaviors, although they were not known as surging before (Gilbert et al., 2018; Jacquemart et al., 2020)."

**L57**: "Recent glacier collapse events on the QTP". Have there been more than the two at Aru? If yes, please specify, otherwise maybe just refer to the Aru cases specifically.

Reply: The sentence now reads: "The detachment of Aru glacier on the QTP has raised concerns on the stability of glaciers there, especially under intense climate warming."

**L59+61**: "glacier instabilities" what kind of instabilities are you referring to here? Just glacier detachments or also ice avalanches, surges etc.? Maybe this can be formulated to be a bit more specific.

Reply: The "glacier instabilities" here include glacier detachments, glacier ice valances and surge

movements. We have rewritten the sentence as follows "While previous studies of glacier ice valances and surge movements on the QTP mainly focused on the Karakorum and West Kunlun mountain regions where a large number of surge-type glaciers exist (Bhambri et al., 2020; Leinss et al., 2019; Yasuda and Furuya, 2015), little is known about glacier instabilities in the inner region of the plateau."

**Figure 1**: It would be nice to show the location of the weather station in this first Figure. The caption suggest that the figure shows something about the geology, when in fact only the topography and the faults ('seismic setting'?). That said, I think a geological overview would be well placed here since the authors later make a reference to fine grained rocks, but never describe the region's geology. Such a description should be added to the Study Site section. Lastly, I am not sure the yellow dots (ice avalanches triggered by the 2001 EQ) add much information here, since they are never mentioned again.

Reply: We have added the weather station shown as a white square in Fig. 1. We have also added some sentences giving a geological overview of the study region in Section 2.1. The yellow dots (The ice avalanches triggered by the 2001 EQ) here indicate that the KLP-37 is located in a region that is venerable to seismic hazard, which was also mentioned in the discussion part. So we would like to keep the yellow dots in Figure 1.

**L104**: please clarify what you mean by "pre-collapse" event

Reply: We have removed the sentence for a clear statement.

**L106:** since you just talked about both the east and the west branches in the previous sentence, it would be helpful to specify relative to which tongues the lakes are.

Reply: The sentence now reads "we can observe an ice-dammed glacier lake in front of the glacier's east branch and a supraglacial pond on the west branch's tongue surface."

**L108:** "We can observe fine debris …" how did you observe this? High resolution images are likely not enough to determine this adequately. If you have not had a chance to investigate in the field, a reference to the geology of the area (L110 "weak bedrock") would be helpful.

Reply: We have rewritten the relevant part. We have cited a reference (see below) in geological descriptions (Section 2.1) to show that the glacier likely lays on a weak bedrock.

Wu, X.H., Qian, F., and Pu Q.Y.: Quaternary Geology of the Eastern Kunlun Mountain, Symposium of Geology on Tibetan Plateau, pp. 1-18. Beijing: Geological Publishing House, 1982. (In Chinese)

**Figure 2:** In this caption, as well as somewhere in the text, please note (and possibly show) the distance between the glacier and the railroad. Also, please label what the yellow polygon indicates.

Reply: We have annotated the distance between the glacier and the railroad in Fig. 2b. The yellow polygon indicates the outline of the tongue of the glacier's west branch, which has been labeled in Fig. 2a.

**L133:** In this paragraph, it is not always clear which DEMs you made and which you just acquired. Did you generate any of the radar data based DEMs yourself (seems like they are all freely available products?). A little bit of clarification would be helpful here.

Reply: We have clarified that the SRTM and HMA DEMs are publicly available, and the commercial TanDEM is provided by the German Aerospace Center under an academic license.

**L155:** define $\partial t$

Reply: We have added that "Δt is the time span of the two images".

**L157:** "which is set to zero because all the optical images used are orthorectified and georeferenced products" is not a satisfying reason for setting that value to zero. In my experience, orthorectified Planet images can exhibit a lot of jitter. Since you have your cross-correlation pipeline running in MicMac, you should be able to get an estimate of this value from the off-glacier cross correlation results.

Reply: Thank you for pointing out this. The sentence now reads: "$\varepsilon_{geo}$ is the relative georeferencing error between the two images; Δt is the time span of the two images. Note we estimated $\varepsilon_{geo}$ for Planet image pairs from off-glacier cross-correlations, and for KH-9–Planet image pair from coordinate differences at the selected ground control points". We have also updated the uncertainty of the velocity estimate using Eq. (1) by considering the georeferencing error between two images.

**Figure 3:** The Nuth and Kääb approach for co-registering DEMs is pretty standard – the image could easily go into the supplementary material.

Reply: We have moved the Fig.3 into the supplementary material as suggested.

**L194**: I'm not sure that I agree with the authors approach to the applying single offsets here. In terms of snow penetration, if I had an estimate of penetration error with elevation, I think I would chose to correct it based on a fitting a curve to the data. The authors state that they are not interested in the high reaches of the glacier, but then show such results in Fig. 6. It would be interesting to see a bit more discussion of the effects of these error sources in the discussion. Do they significantly change the amount of thinning/thickening that we can see on the different parts of the glacier?

Reply: We have added a figure showing the elevation difference between the C-band and X-band DEMs over the KLP-37 glacier in the Supplementary file (Fig. S3). It's true that the penetration depth varies with the changes in elevation. Following the good suggestion, we have implemented a curve fitting to the elevation difference and then applied the correction.

We have also added some sentences in Section 4.2 to discuss the influence of penetration correction on the estimation of glacier thickness change.

**L215:** I don't understand what you mean by "standard deviation of the mean elevation change of its elevation band"? Aren't you just using the standard deviation of each elevation band (as indicated in equation 3?).

Reply: We have rewritten the sentence as follows "We assumed that the error for each pixel of elevation change ($\varepsilon_{\Delta h}^{i}$) is equal to the standard deviation of each elevation band".

**L223:** Please add a reference to the least-squares fit … isotropic variogram method, or describe in more detail.

Reply: We have added two references (i.e., Wang and Kääb, 2015; Magnússon et al., 2016) here.

**L231:** Why only three pairs? Processing all possible pairs would make your results more robust.

Reply: Following your suggestion, we have redone the image correlation of Planet images. We have shown correlation results for seven image pairs spanning between 2009 and 2020, with each image pair having a time span of about one year. The analysis and interpretation of the correlation results were also updated (Section 4.3).

**L251:** It is not clear to me what "The snowpack downstream of the crevasses collapsed" means. Was

there an avalanche? Or just different snow conditions over a crevassed area? Or a change to the actual glacier ice? Please clarify this point.

Reply: Sorry for the unclear statement. The sentence now reads: "Satellite images show that the transverse crevasses in the glacier cirque had commenced in 1975 and became more evident in the following years (see the red arrows in Fig. S2). Specifically, the crevasses' length and width increased apparently after 2013."

**L254:** If it stayed unstable for four decades, does that mean it is actually stable? Clarify the point you are trying to make with this statement.

Reply: We have rewritten the sentence as follows "The highly developed crevasses in the glacier accumulation region since the 1970s and the widening of the crevasses in recent years indicate that the glacier may develop towards destabilization".

**Figure 4:** It is very hard to see much in this image, but there is a lot of empty space! I suggest rotating the image and making it 3x4 or 4x3 cells, so that it is easier to see something.

Reply: We have modified Fig. 4. Two Planet images acquired on 2020/08/29 and 2021/03/21 have also been added in Fig. 4. Note that the figure is numbered as Fig. 3 as we have moved the old Fig. 3 into the supplementary file.

**L267:** Did you determine the uncertainties of the area in the same way that you did for the advance velocities, or some other method?

Reply: The lake was independently delineated five times, from which we estimated the uncertainty of the lake area. We have clarified this in the revised manuscript.

**Figure 5:** It would be really nice if each dot also had a horizontal bar that indicated over what time span the velocity was calculated.

Reply: Good suggestion. We have modified the figure by adding a horizontal bar for each velocity estimate.

**L279:** Clarify that you DON'T mean that the velocities were similar between 2009–2015 and 2015–2019, but instead were stable during those periods, but then jumped between 2015 and 2016.

Reply: The sentence now reads: "The velocities were stable during the two periods 2009–2015 and 2015–2019, but jumped between 2015 and 2016."

**Figure 6:** Please make the colorbars a little bit larger and use the same scale for each image (in which case you can get away with just one colorbar for each line of plots). It is very hard to interpret the changes when the scales change.

Reply: We have enlarged the colorbar of the figure and set the a same colorbar for all the sub-images.

**L295:** How much could this assessment be influenced by the radar penetration depths?

Reply: The KH-9–SRTM and SRTM–HMA10 DEM pairs are influenced by radar penetration depths. To infer how much the radar penetration depths will impact the elevation change estimates, we have calculated the elevation differences between the X-band and C-band SRTM at point ′T′ and obtained an elevation difference of 2.44 m. Considering that we have applied a correction of 2.13 m (given the correction function of $y=0.0046×h-21.5335$ and the elevation of 5145 m at the point ′T′) to the SRTM DEM, a residual of 0.31 m would remain in the elevation change estimate. Thus, we infer that the elevation change errors due to SRTM penetration are about 0.01 m·a$^{-1}$ and 0.03 m·a$^{-1}$ for the KH-

9–SRTM and SRTM–HMA10 DEM pairs, respectively.

**L298:** over the whole east branch? Or another window? Please clarify.

Reply: The mean elevation changes were calculated for the whole east branch. We have clarified this in the revised paper.

**L299:** did the east branch only thin or also retreat?

Reply: The east branch did not show retreat behavior in the past 40 years. We have clarified this in the revised paper.

**L305:** I'm not sure I agree with the term *bulging* for what is described here (and depicted in Figure 7b): Bulging implies that there was an upward motion of ice surface, but really there is just advance of ice. The DEM difference is mostly against previously un-glacierized terrain, so obviously this appears as thickening in the DEM difference, but it does not reflect thickening of ice in a place where the was ice previously. That said, I do see bulging Figure 6, but am having a hard time finding that in the cross-section.

Reply: The word "bulging" here indeed may lead to misunderstanding. We have replaced "bulging" in Fig. 7b with "Increasing elevation". We also modify the relevant sentences in the text to avoid the use of "bulging" to indicate elevation increasing due to the advance of glacier toward the place where there was no ice previously.

**L306:** What makes it not a surge, but only surge-like? This could be an interesting topic to elaborate on in the discussion.

Reply: Surge-type glaciers periodically alternate between long periods of slow flow (the quiescent phase) and short periods of fast flow (the surge phase). Surging yields down-glacier transport of mass and often results in large and sudden glacier advances. Here we use the word "surge-like" because the flow velocity of KLP-37 shown stepwise increase between 1975 and 2016. This kind of flow behavior resembles the surging movement of surge-type glaciers, which has also been reported on the Aru or Amney Machen glaciers that have detached in the Tibet Plateau. However, we cannot conclude that the KLP-37 glacier is a surge-type because we have not captured the periodical surge advance signal. We have elaborate on this in the revised text (Section 4.2).

**Figure 7 :** Can you indicate the years next to the different DEMs? That would help interpret the changes. Then, it would be nice if the colors were on a gradient scale along age, so that it is visually clear which way the changes progressed.

Reply: Good suggestion. We have modified Fig. 7 by adding the timespan for each DEM pair and by changing the color scale for each line.

**Figure 8:** Because the x-axis runs from upglacier to downglacier in the previous plots, I think it would be beneficial to flip the x-axis in this plot (so that higher elevations are on the left and lower elevations are on the right).

Reply: Good suggestion. We have modified Fig.8 as suggested.

**L334:** Over what area where these maximum velocities measured? Single points?

Reply: The maximum velocity over each period represents the maximum value within the glacier tongue area, and the value was measured on one point (pixel). We have clarified this in the revised version.

**Figure 9:** Please use colorbars that are not divergent (the blue to white to red implies that blue has an

opposite signal to the red) but rather continuous, and use the same scale in each plot. Additionally, it would be very nice to see the velocities outside the glacier as well. Is there actually enough accuracy to distinguish the glacier from the surrounding landscape? Lastly, you keep referring to the 4800m elevation line, so it would be very nice to show that line in the plots.

Reply: We have changed the colorbar of Fig. 9 to a convergent one. The velocities outside the glacier were illustrated as well. It can be seen from Fig. 9 that the flow of the glacier can be distinguished from the surrounding area for velocity field after 2016, especially at the glacier front region. We also added the elevation contour line of 4800 m in the revised version.

**L349:** Your surface flow velocities and terminus advance velocity differ by two orders of magnitude. I don't see the explanations you offer as plausible, but suspect an error in the processing of the cross-correlation.

Reply: Thanks a lot for pointing out this. We have redone the image cross-correlation with the Planet images and updated the velocity maps and the relevant analysis in the revised paper(see Section 4.3).

**L370:** How did you (or Wu et al?) determine the lower permafrost altitude in the field? It would also be good to take a look at the two global permafrost maps (Gruber et al., 2012, Obu et al., 2019), and reference those.

Reply: Wu et al. (2005) used a 50-MHz ground-penetrating radar to detect the boundary of permafrost in the study region, and they interpreted the lower limit of permafrost from nine radar profiles to determine the boundary between the permafrost and non-permafrost region. We also checked the permafrost map based on TTOP modeling (Obu et al. 2019) and found that the permafrost probability value over the KLP-37 glacier tongue region is 1. We have clarified a bit on the lower permafrost altitude of the study area in the revised paper (see Section 5.1).

**L374:** If my memory serves me correctly, Leñas glacier was deemed temperate… but I think something similar was found at Flat Creek. Maybe double check the references?

Reply: From the prolonged preservation of the collapse deposits on the Leñas glacier, Falaschi et al. (2019) suggested a potentially cold ground temperature regime for parts of the Leñas glacier and forefield. For the Flat Creek glacier, Jacquemart et al. (2020) concluded that it was polythermal, with a thin cold-ice tongue. We have added some clarifications in the revised version.

**L379:** The statement that the ice-dammed lake exerts a hydrological influence on the glacier tongue needs to be backed up significantly. What kind of influence? This seems like it contradicts the cold-ice edge that is keeping the lake locked in. If the lake has not decreased in size, I don't see how it could influence the dynamics of the glacier in any substantial way.

Reply: We have removed the relevant statement in the revised paper.

**L384:** I don't understand what you mean by "deflection region"

Reply: Sorry for the unclear description. The "deflection region" represents the place where the flow direction of the glacier changes. We have rewritten the sentence for clarity in the revised paper.

**L386:** It seems to me that the narrowing of the topography which kind of "pinches" the glacier tongue would help stabilize it, rather than making it unstable. Please clarify how you think this factor contributes to a destabilization.

Reply: The topography of the KLP-37 glacier contributes to destabilization from two aspects. First, the ice flow direction changes at the elevation of about 4800 m due to the local topography.

Therefore, large ice masses accumulate above 4800 m, and the driving pressure thus increases. Second, the slope angle is about ~10° above 4800 m but increases to ~20° at the lower place. The steep topography provides a preconditioning factor for the unstable status of the glacier tongue. We have clarified these two aspects in the revised paper.

**Figure 11:** Why are you only showing summer temperatures? At what elevation are these temperatures? The current graph does not support your claim that the area is in permafrost.

Reply: We have modified the figure and shown mean annual air temperature instead of mean summer temperature at the Wudaoliang meteorological station. The elevation of the weather station is 4613 m, which has been remarked in the caption of the caption of Fig. 11.

**L395:** I am a bit hesitant about this interpretation of the influence of temperature and precipitation on the flow speeds. Firstly, what exactly are you referring to by "flow velocity"? The advance of the glacier tongue or the mapped velocities on all of the tongue? This needs to be clarified. For the advance of the glacier tongue, the change really just happened between 2015 and 2016, but was likely the result of accumulation that happened further upstream in the years before. Please elaborate on these points.

Reply: We have clarified in the revised paper that the "flow velocity" refers to the mapped velocities on all of the tongue from image cross-correlation. Note the maximum flow velocity from our updated processing are comparable to the snout advance velocities.

We agree that the abrupt velocity increase between 2015 and 2016 could likely be the result of accumulation that happened further upstream in the previous years. We have removed the statements that the acceleration after 2015 may be induced by the high temperature and precipitation amount during 2015 and 2018. The relevant sentence now reads: "Given the stepwise increase of snout advance velocity between 1975 and 2021 and the more than tripled mean flow velocity below 4800 m between 2009 and 2020 (Section 4), we suggest that the climate warming in the study region likely contribute to the long-term acceleration of the glacier tongue."

**Figure 12:** Why are you only considering the tongue in you hazard assessment? The crevasses seem to be much higher on the glacier, and the situation at Aru showed that that was where the detachment initiated. At the very least, I believe that it would be valuable to run the model with the two endmember volumes.

Reply: Thank you very much for your good suggestion. We have presented avalanche modeling including two endmember scenarios with different source volumes (Section 4.4).

**L435:** Can you get that much advance by just internal deformation? I don't know the answer, but feeling like there might have to be some sliding, at least in the fastest parts.

Reply: We have clarified this in Section 5.1 in the revised paper.

"The cross-correlations of the 2019–2020 image pair show that the mean velocities along the central profile were about 9.8±1.4 and 22.3±3.2 m· $a^{-1}$ for regions above and below 4800 m, respectively, corresponding flow velocities of 2.7±0.4 and 6.1±0.9 cm·$d^{-1}$. These values are on the same order of magnitude as the estimated velocity of 2 cm·$d^{-1}$ for pure ice deformation (Leinss et al., 2019; Round et al., 2017). We thus suggest that internal ice creeping should mainly account for the downslope movement of KLP-37. However, given the fast and homogenous flow in the lower part of the glacier tongue, it is reasonable to postulate that basal sliding is partly responsible for the flow dynamics."

**L458:** I don't see how the current glacier flow velocities/basal friction parameters influence how far the mass flow resulting from a glacier detachment can travel. If the entire thing detaches, basal friction decreases to essentially nothing, and that is what determines the runout distance.

Reply: We agree that it is unreasonable to link the current glacier flow velocity to the possible runout distance of a glacier. We have removed the relevant statement.

**L461:** I am not convinced by your justification for selecting the moderate friction parameters. I am not sure exactly what findings you are referring to with reference to the Allen 2009 paper, but until very recently, we had hardly heard about glacier detachments, and it is unlikely that a comprehensive assessment of basal friction values was published over 10 years ago. Furthermore, I am not aware of any findings that link the basal shear stress during regular glacier flow to the runout distances of glacier detachments, most certainly not for making a difference between surging and non-surging glaciers. Please clarify substantially.

Reply: Here, we try to use the Voellmy-Salm model to investigate the runout distance of potential detachment on KLP-37. Selecting proper friction parameters (i.e., coulomb-type friction $\mu$ and turbulent friction $\xi$) is critical for running the simulation. Because it is impossible to obtain these friction parameters directly, we empirically determine the ranges of $\mu$ and $\xi$ from previous studies. Although the assessments of these friction values for glacier detachment events are rare, simulations of previous ice/rock avalanches using the VS model can give us clues.

Allen et al. (2009) implemented VS modeling to reconstruct past eight ice/rock avalanche events from different glacial environments and summarized the best-fit friction parameters that can obtain flow paths fitting well with realities. The eight simulated events include unequal fractions of ice and water content and extremely variable topography. The best-fit values of $\mu$ and $\xi$ for these events range between 0.05~0.2 and 1000~4000 $m \cdot s^{-2}$, respectively. Retrospective analyses of a few glacier detachment events have also revealed friction parameters laying in the above ranges. For example, the best-fit $\mu$ for the first and second Aru glacier detachments are 0.11 and 0.14 (Kääb et al., 2018); The 2002 Kolka detachment has best-fit values of 0.05 and 2700 $m \cdot s^{-2}$ for $\mu$ and $\xi$, respectively (Allen et al. 2009).

We have clarified why we chose the moderate friction parameters 0.15 and 2500 $m \cdot s^{-2}$ for the VS modeling in the revised paper (Section 3.4). We have also compared the runout distance from VS modeling with the simple calculation from the angle of reach ("Fahrböschung") (Section 4.4). We have removed the statement that relates glacier flow velocity to the possible runout distance estimate.

**L474:** I don't understand what you mean by "experiencing ice-rock collapses"?

Reply: The sentence now reads :"Although plenty of glaciers in QTP have been retreating in the last several decades, glacier advancing has been ubiquitously observed either on surge-type glaciers or on those where ice-rock avalanches have occurred."

**L475:** I don't follow how Kääb 2020 suggest that any glacier advance is an indicator for a glacier instability (though I suppose this depends on how you define glacier instability – not all are hazardous).

Reply: To avoid misunderstanding, we have rewritten the sentence and now it reads, "Kääb et al. (2021a) suggested that glacier detachments could be seen as extreme endmembers of the range of surge-type and surge-like glacier instabilities (2020), supported by the fact that many of glacier detachments exhibited surge-like advance ahead of the failures or occurred on surge-type glaciers

themselves."

**L481:** Please back up the statement of "the ice-dammed lake influencing the dynamics of the glacier tongue of the west branch". This is currently not supported by any of the data, nor has any relationship between the two been mentioned prior to this.

Reply: Thanks for the reminder. We have deleted the relevant statements.

**L487:** Please back up the statement "was transported from the upper region by a historical seismic event". How did you determine this to be a logical possibility? Is there evidence of something like this elsewhere in the region? So far, we have not seen earthquakes causing detachments of low-angle valley glaciers… (not saying it's not possible though…).

Reply: Thanks a lot for your reminder. We have deleted the relevant statements.

**L492:** In addition to ground based InSAR, optical camera based systems are probably cheaper and similarly effective, if it's just for monitoring.

Reply: The sentence has been modified to "In the future, monitoring techniques with short temporal sampling rates such as ground-based SAR and optical camera-based systems should be employed to capture the transient or accelerating signals of surface motion".

**L494:** I don't understand what you mean by saying "our simulations provide an alternative solution for assessing the hazard of an impending glacier collapse". Alternative to what? Is the glacier detachment impending, or just a possibility (I would argue that it is probably more likely that nothing will happen).

Reply: We have revised the sentence to make it more clear, and now it reads: "Our simulations of the runout extent using avalanche modeling, combined with the empirical estimates of runout distance using the angle of reach, provide a preliminary assessment of the hazard influence of a potential glacier detachment."

**Technical corrections**

Terminology: Over recent months, the term *glacier detachment* seems to have become the term of choice for describing the catastrophic detachment of low-angle valley glaciers. Rather than using the somewhat fuzzy term "collapse", which is frequently used for rock avalanches, slope failures etc., I recommend changing the terminology to glacier detachment throughout the manuscript. Example: Due to hazardous threats of glacier  detachments to … (Line 37).

Reply: We have changed the terminology "glacier collapse" to "glacier detachment" throughout the manuscript.

Use of tenses: There are quite a few improper uses of past tenses. I have noted these below where I caught them, but am likely to have missed a few. In general, I suggest to put anything that the authors have done in the past tense. Example: We chose a set of parameters, we analyzed these images etc.

Reply: Thanks a lot for the reminder. We have checked the tenses for verbs throughout the paper.

If your line numbers have changed, I also have these edits in a document (hand written). Please let me know via the editorial office if having this document would be helpful.

L26: *tripled* not trebled ; L39: *glacier* not glacial; L40: *was* not have; L41: *was* not is; L46: *two* not

twice; L46: remove *the lower parts* ⟶the entire glacier was implicated in 2015; L53: *on* or *from* not in; L59: *mean global rate*, not global mean rate; L67: …identified the glacier, *which is* close to …; L68: intense *crevassing* on the glacier surface *raised* the question *whether* a hazardous ice avalanche *might be* imminent.; L71: *past* not recent L75: *discuss* not discussed; L76: *estimate* not estimated; L77: *occurred* not occurs; *discuss* not discussed;

Reply: Corrected.

L104: the west branch's *terminus lies about 220m lower* (Fig. 2a); remove "Particularly"

Reply: Done.

L112: *overlayed* not overlapped; *The* ice-dammed lake and supraglacial pond are also *shown.*

Reply: Done.

L118: Why are Table 1 and Table 2 shown at the end of the document? It would be much nicer to have them where they are referenced.

Reply: We have placed Table 1 and Table 2 into the main text.

L121**:** I would say that glacier changes are *often* mapped on Landsat etc. images (partly bc for decades that is the best we had) and that they have a *resolution of 10-30m.*

Reply: The sentence now reads: "Glaciers changes are often mapped on the freely available Landsat and ASTER satellite images, which commonly have a resolution of 10–30 m (Bolch et al., 2011; Scherler et al., 2011)"

L123: *used* (if you got it free) or *acquired* (if you bought the images) not "collected" (collected is used primarily for data collection that you do in the field). This comes up several times in the manuscript.

Reply: We have replaced the word "collect" to "use" or "acquire" according the suggestion.

L127: Just say …*spy satellite to reconstruct the topography of KLP-37 in 1975.*

Reply: Done.

L161: *generate* DEM*s from* KH-9 stereo images

Reply: Done.

L177: *to* a spatial *posting* of; The DEM pairs *need* to be …

Reply: Done.

L185: *posting* or *resolution* not post

Reply: Done.

L196: odd use of while

Reply: We have removed the sentence containing the word "while".

L200: similar to  previous

Reply: Done.

L248: insert space after KH-9; *past* instead of recent

Reply: Done

L249**:** *A time-lapse* of optical images …

Reply: Done

L251: Remove "KH-9 image" since the crevassing didn't happen in the image, but rather in the world. The images just show it.

Reply: Done

L262: *is* instead of were

Reply: Done.

L277: *the past* instead of recent

Reply:  Done.

L278: *and* instead of while; *rose to more* instead of had been higher

Reply: Done.

L341: *tripled* not trebled

Reply: Done.

L350: remove one *not*

Reply: We have written the relevant part.

L360: *advanced continuously* instead of "had been progressively advancing"

Reply:  Done.

L361: *accelerated* instead of was accelerating

Reply: Done.

L362: remove *temporal*

Reply: Done.

L363: …conditions of *the glacier,* topography …

Reply:  Done.

L371: *terminus* instead of termini

Reply: Done.

L377: … crevasses *to reach the* sliding surface …

Reply: We have written the relevant part.

L403: *dense* is an odd choice of word here – what do you mean?

Reply: We have replaced the word "dense precipitation" to "heavy precipitation".

L403: *was* instead of "has been"; I don't understand what you mean by "within 40 days before"? 40 days prior? Or during the 40 days prior? Or up to 40 days prior?

Reply: We have rewritten the sentence as follows "Specifically, heavy precipitation accounting for about 88% of the total precipitation of 2016 was recorded during the 40 days prior to the Aru glacier collapse".

L501: with *a total* advance instead of "with an accumulative distance"

Reply: Done.

---

## Author Response (AR2)

**Reply to the Comments**

We would like to thank the two referees for their further suggestions and constructive comments, which helped to improve the paper. The changes in the manuscript are highlighted in blue color. Our item-by-item responses to the comments are provided below.

**Reviewer #1**

The authors did substantially revise the paper. The paper is better structured now, and unreasonable elements of the results are revised.

Two final suggestions:

Line 281: "indicate that the glacier may develop towards destabilization". As you don't know if the glacier will destabilize, I suggest something like "that the glacier may develop into a less stable regime."

Reply: We have modified the text following the suggestion.

Line 395: "getting more active towards destabilization". Similar to my above comment to line 281.

Reply: We have done the modification.

**Reviewer #2**

L14: rephrase to say that "few opportunities have presented themselves to assess the potential hazards of a glacier prone to detachment" or something similar.

Reply: The sentence has been modified following the suggestion.

L26: Feels like a jump from the information you have provided in the abstract so far. Can you clarify this statement?

Reply: The sentence now reads "With a combined analysis of the geomorphic, climatic, and hydrologic conditions of the glacier, we suggest that the flow of the glacier tongue is mainly controlled by the glacier geometry, while the presence of an ice-dammed lake and a supraglacial pond implies a hydrological influence as well."

L 36: tens of km seems a bit on the high end

Reply: The words have been modified to "more than ten kilometers".

L37: typical volumes = $10^6 - 10^7$, full range would be $10^6$ to $10^8$

Reply: Corrected.

L86: what do you mean by sand slate? Slate is a clayey rock, I don't think something like sand slate exists.

Reply: We have modified the relevant part, please see Line 85-91.

L86: What are these grains? Sandstone and slate probably do not have grains with diameters of 2-5 cm. That would be a conglomerate. Are you talking about sediment found in the region? That is pretty large… please clarify.

Reply: We have clarified that the glacier lies on a sequence of glacial deposits, Wangkun till. The till is mixed breccia composed of angular, poorly sorted gravels of slate, meta-sandstones, and mudstones (Song et al., 2005).

L87: What do you mean by rock fragments that are filled with fine-grained sediments?

Reply: The sentence has been modified to avoid misunderstanding.

L140: here and everywhere else, change TanDEM to TanDEM-X DEM (TanDEM-X is the satellite, TanDEM-X DEM is the dem generated from the satellites data).

Reply: Done.

L149: higher accuracy than HMA DEM?

Reply: The vertical accuracy of HMA DEM was reported with an accuracy of 3.01 m in the Karakoram region, Tibet Plateau (Kumar et al., 2020). However, the TanDEM-X DEM has a nominal accuracy of 2 m in vertical direction. We have added a cation (Kumar et al., 2020) here.

L198: The figure in the supplementary material shows huge variations in the data. Would be nice to add this information here for a bit more context…

Reply: We have added a sentence at Line 202: "Huge variations of the elevation differences can also be observed (Fig. S3), highlighting the heterogenous penetration differences between the two SRTM DEMs."

L208: it's not clear to me how you came up with this empirical value…

Reply: We have clarified that the value was estimated from the seasonal snow depth in the KLP region (Tian et al., 2014).

L282: What do you make of the "swollen body" as you call it? Just the healthy state of the glacier during that time? Evidence of an instability? A little bit more explanation would be useful.

Reply: The satellite images show that the glacier tongue has a swollen body in 1975, while the body becomes flat in 2021. This change indicates the continued ice loss of the tongue in the past decades, and the glacier is in a negative mass balance state. We have explained this in the revised paper.

L290: "The lake peaked in summer and decreased in winter" -> is this always the case or just once? Formulate statement to clarify this.

Reply: The sentence now reads "The lake area in summer was commonly larger than that in winter. The area larger than 10000 $m^2$ all occurred during July and September."

L312: What is going on in the ASTER DEM difference (Fig. S6)? There seem to be large, non-random offsets between this and the TanDEM-X DEM… I'm not sure these results could be trusted…

Reply: It is true that the ASTER DEM has large errors. However, the large errors are mainly distributed near the mountain ridges and steep slopes, which thus would not remarkably affect the elevation differences over the KLP-37 glacier. We have clarified this in the revised paper, please see Line 324-325 and Line 331.

L321: Delete sentence "Note the estimate…" You've already stated that this is problem, and presented your solution. I think you could even move some of the explanation of how you derive the influence on the measurements into the methods section and then only provide the results here.

Reply: We have deleted the sentence as suggested. The relevant words have also been rewritten.

Fig5: I find it counter-intuitive that blue = elevation decrease and yellow/red = elevation increase. Also, it would be very helpful if you could put the dates of the two DEMs used for every results figure into the figure panel directly, so that the reader does not have to refer to the text to remember the dates.

Reply: The color of Fig. 5 has been changed by using the red color representing elevation decrease. The date of each DEM has also been labeled.

Fig 7: Why are there gray bars between the tongue and the cirque region that are not attributed to either?

Reply: The yellow, cyan, and green bars correspond to the elevation ranges of glacier cirque, transfer zone, and glacier tongue as shown in Fig. 6, respectively. We have added this description in the caption of Fig. 7.

L400: there are two Aru Glaciers (Aru-1 and Aru-2 in Kääb et al., 2018). Change the wording in the paragraph to reflect this or specify which Aru glacier you are referring to.

Reply: We have specified the Aru-1 or Aru-2 glacier when referring to the Aru glacier.

L403: I don't see how the gravitational potential energy is a function of the geometry of the glacier, but rather a function of the height and mass of any given point. Clarify why the geometry is relevant in this context.

Reply: The sentence now reads "the glaciers' widths gradually narrowed from the source region to the tongue, thus resulting in a large amount of ice mass accumulating at the glacier front and further leading to large gravitational potential energy there."

L406: How did you estimate this volume? Are the two decimals justified for this estimate?

Reply: We estimated the ice volumes for the two scenarios based on the ice thickness map derived by Farinotti et al. (2019), who provided an ensemble-based estimate for the ice thickness distribution of all glaciers included in the Randolph Glacier Inventory (RGI) apart from the Greenland and Antarctic ice sheets (See Method Section 3.4, Page 11). We have clarified that the increase of decimals for the

volume values would not affect the runout distance estimates significantly.

Fig. 10: label legend, don't put label over the content of the figure. Also, why does part of the glacier not have a thickness estimate, but the tongue is wider below it?

Reply: The ice thinness map of Farinotti et al. (2019) was derived based on the Randolph Glacier Inventory (RGI). However, the outline of KLP-37 from RGI does not include the glacier tongue region KLP-37. We have clarified this in the Method (Section 3.4) and in the caption of Fig. 10.

L433: What do you mean by "the landform"? Do you just mean the glacier tongue?

Reply: We have modified it into "the glacier front landform" to avoid misguide.

L437: very complicated way of saying that a global permafrost map classifies the area as permafrost with very high likelihood. I suggest rephrasing, because the reader initially thinks that you mapped permafrost yourself.

Reply: The sentence now reads "Global permafrost extent mapping by Obu et al. (2019) also classifies the glacier font area as permafrost with very high likelihood".

Section 5.1: I am a bit confused by this part of the discussion, because you introduced this as the glacier tongue throughout the paper… I think you need the rephrase this part of the discussion slightly to explain why one could also consider this something that is not part of the glacier. I would say try to make this fairly short and limit to most important key factors only. Otherwise it takes away from the main message of the paper.

Reply: We have rewritten the paragraph to make it more clear. The relevant explanations have also been shorted.

L466: I think there are better papers to cite, or at least additional ones here, especially Faillettaz et al. between 2011 and 2016 and Pralong and Funk around 2005/2006.

Reply: The references have been replaced to Faillettaz et al., (2015) and Pralong and Funk (2005).

L486: Can you see any evidence of upstream ice accumulation in the DEM differences?

Reply: We have modified the sentence to: "The abrupt velocity increase below 4800 m between 2015 and 2016 could likely be the result of mass accumulation, as evidenced by apparent elevation increase above 4700 m from the DEM difference between 2010 and 2014 (Fig. 5g)".

L525: Which publication are you referring to here? There is no Kääb 2020 in your bibliography…

Reply: The citation has been modified to Kääb et al. (2021a).

L544: I don't understand what you mean with the sentence about adding to the diversity. Please clarify.

Reply: We have added some explanations in the relevant paragraph.

L545: You mention the geometry of the glacier throughout the paper, but the point that you are trying to make hast not become entirely clear to me yet. It seems that you are suggesting that the particular

geometry of this glacier makes it more prone to detaching, but it seems to me that the narrowing at the tongue rather provides important buttressing. However you see this, I think it would be worth stating your points a bit more explicitly.

Reply: Thanks for the reminder. We agree that the narrowing at the tongue could provide a buttressing effect for the upstream ice mass. However, on the other hand, the specific shape also prevents the glacier from adjusting its geometry to the changed driving stresses with the accumulation of ice mass. This thus continuously increases the stresses on the frozen terminus and margins, until reaching a critical point where the resisting force is eventually overcome and an acceleration or detachment occurs. We have clarified these two effects in the revised paper (Line 488-492).

L550: Not really three-dimensional, rather one type of 2-D (horizontal) and one type of 1-D (vertical), so I think it is better to just say horizontal and vertical.

Reply: The words have been revised.

L553: It would be interesting to hear you assessment of what critical signs of further destabilization could be. Up-glacier growth of the fast-moving zone? Additional crevassing? Further acceleration of the front? Appearance of shear margins along the edges?

Reply: Good suggestion. We have added a sentence here, and please see Line 567-569.

"Particular attention should be paid to the critical signs pointing to further destabilization, such as the further acceleration of the glacier front, up-glacier growth of the fast-moving zone, additional surface crevassing, and appearance of shear margins along the edges."

L567: Change in flow direction – also a point that I'm not sure how it plays into the dynamics. Similar to the V-shaped geometry, I think it's worth clarifying throughout the manuscript why you think that this is important (other than the fact that the Aru glaciers also had a bit of a bend in them).

Reply: The sentence now reads "The change of flow direction of the glacier at an elevation of about 4800 m due to the local topography, coupled with the "V" shape of the glacier tongue geometry, presumably leads to large ice mass and stress accumulating at the glacier front and plays a crucial role for the surge-like behavior". We have also added some clarifications at Line 488-492 and Line 558.

L568: how does the hydrology influence the tongue? Do you have any evidence for this happening? Otherwise maybe make this statement more carefully? This topic was not brought up again during the discussion, so it comes as a surprise in the conclusion because I don't think you have adequately made this point. It's fine to say that hydrology may play a role but that the mechanisms of it are not clear, but I think you need to be a bit more specific.

Reply: We have added some clarifications in the discussion part (Section 5.2, Line 510-515) to show how does the hydrology may influence the tongue dynamics. We also clarified that the mechanisms of the hydrological effects are not clear in the Conclusion part (Line 585).